# Optimal Aggregation of LLM and PRM Signals for Efficient Test-Time Scaling

**Peng Kuang**[1]**, Yanli Wang**[2]**, Xiaoyu Han**[3]**, Yaowenqi Liu**[3]**, Kaidi Xu**[4]**, Haohan Wang**[3]

[1]Zhejiang University, [2]Imperial College London, [3]University of Illinois Urbana-Champaign
[4]Drexel University
pengkuang@zju.edu.cn, kx46@drexel.edu, haohanw@illinois.edu

## Abstract

Process reward models (PRMs) are a cornerstone of test-time scaling (TTS), as it is designed to verify and select the best responses from large language models (LLMs), significantly improving the performances. However, this promise is challenged by recent benchmarks where simple majority voting, which ignores PRM signals, occasionally outperforms standard PRM-based selection. This raises a critical question: How can we effectively utilize verification signals from PRMs for TTS? To address this, we start by developing a theoretical framework for optimally combining signals from both the LLM and the PRM. Our framework reveals that the optimal strategy is a weighted aggregation of responses, a strategy whose effectiveness hinges on estimating weights that capture the complex interplay between the models. Based on our theoretical results, we empirically show that these optimal weighting functions differ significantly across LLM-PRM pairs and, notably, often assign substantial negative weights. Motivated by these insights, we propose efficient pre-computation methods to calibrate these weighting functions. Extensive experiments across 5 LLMs and 7 PRMs demonstrate that our calibration method significantly boosts the TTS efficiency, surpassing the performance of vanilla weighted majority voting while using only 21.3% of the computation. Ultimately, our work demonstrates that investing in a more intelligent aggregation strategy can be a more convincing path to performance gains than simply scaling test-time computation.

## 1 Introduction

The pursuit of advanced reasoning in Large Language Models (LLMs) has largely been driven by scaling up model size and training data (Ouyang et al., 2022). While effective, this approach entails prohibitive computational costs (Snell et al., 2025). An increasingly popular alternative is Test-Time Scaling (TTS)(Liu et al., 2025; Madaan et al., 2023), a paradigm that enhances the performance of a fixed LLM by allocating more computational resources at inference time. A prominent TTS strategy involves generating a multitude of candidate solutions and then selecting the most promising one. This "generate-and-select" framework relies heavily on the quality of the selection mechanism, which is tasked with identifying the correct response from a pool of diverse, model-generated outputs. The central challenge, therefore, lies in designing a selection strategy that can effectively harness the collective evidence from multiple generated responses to maximize final performance.

To address this selection problem, a common approach is to employ a Process Reward Model (PRM) (Lightman et al., 2024; Li & Li, 2025; Zheng et al., 2024), a sophisticated verifier trained on human feedback to score the quality of reasoning steps. The standard protocol, Best-of-N (BoN), simply selects the answer from the single response that receives the highest PRM score. Intuitively, this should leverage the detailed, step-by-step evaluation capabilities of the PRM. However, this intuition is challenged by a surprising and counter-intuitive empirical reality: on recent benchmarks (Zhang et al., 2025b), the far simpler method of majority voting (Wang et al., 2023), which completely ignores the expensive PRM and relies solely on the consensus of the LLM's own generations, can outperform PRM-guided BoN. This paradox suggests a fundamental misalignment in how we utilize verifier signals. If a powerful, costly-to-train PRM can be bested by a simple vote count, it implies we are failing to properly integrate its nuanced feedback.

In this work, we dive into the interactions between LLMs and PRMs to find better aggregation of signals from both models for more efficient TTS. We begin by formalizing the task of aggregating responses as a Maximum a Posteriori (MAP) estimation problem, revealing that the optimal aggregation strategy is not to simply pick the best-scoring response, but to perform a weighted majority vote. Interestingly, the optimal weight for each response is a function of two distinct components: a term derived from the PRM's score, reflecting the quality of the reasoning, and a term derived from the LLM's own reliability. This formulation provides a principled framework for unifying the evidence from both the generator and the verifier.

To understand the practical implications of this theoretical result, we conduct an empirical analysis to characterize the optimal weighting function, uncovering two critical insights. First, the shape of the optimal function is highly dependent on the specific LLM-PRM pair, indicating that a one-size-fits-all approach is inherently suboptimal. Second, we find that optimal functions consistently assign *negative* weights to responses with low PRM scores. This reveals a key deficiency in existing methods (Wang et al., 2024): they fail to leverage the negative evidence provided by a low-quality response. A response judged to be poor by the PRM should not simply be ignored; it should actively count *against* its proposed answer.

Motivated by these insights, we introduce simple yet effective calibration methods to learn approximations of these optimal weighting functions from a small, one-time pre-computed dataset. We propose both non-parametric and parametric approaches that explicitly capture the model-specific nature of the weights and incorporate the mechanism of penalizing low-quality responses. Extensive experiments across 5 different LLMs and 7 PRMs on the MATH (Hendrycks et al., 2021) datasets demonstrate the superiority of our approach. Our calibrated weighted voting method consistently outperforms baselines, including standard BoN and vanilla weighted voting. Notably, it achieves higher accuracy than these methods while using approximately 37.1% and 21.3% of the test-time computation, demonstrating a significant improvement in TTS efficiency. In summary, our contributions are:

- We develop a theoretical framework for optimally aggregating LLM generations and PRM scores, demonstrating that the solution is a weighted majority vote combining signals from both models.
- We empirically characterize the optimal weighting function, revealing its model-dependent nature and, interestingly, the importance of assigning negative weights to low-quality responses.
- We propose practical calibration methods to learn these weighting functions, enabling efficient and effective test-time scaling.
- Through extensive experiments, we show that our calibrated aggregation strategy significantly improves TTS efficiency, achieving superior performance with substantially less computational overhead.

## 2 RELATED WORK

**Test-Time Scaling.** The pursuit of improved model performance without retraining has led to the paradigm of Test-Time Scaling (TTS), which allocates more computational resources at inference time Zhang et al. (2025a). A dominant strategy within TTS is the "generate-and-select" framework, often formalized as Best-of-N (BoN) sampling, where $N$ candidate solutions are generated and a selection mechanism chooses the best one Ichihara et al. (2025). A foundational method in this area is Self-Consistency (SC), which samples multiple diverse reasoning paths and selecting the final answer via a simple majority vote Wang et al. (2023). The intuition is that an answer derived from multiple independent lines of thought is more likely to be correct. While effective, SC's primary drawback is its high computational cost. This has motivated more efficient variations, such as Confidence-Informed Self-Consistency (CISC), which introduced a weighted majority vote based on the model's self-assessed confidence to reduce the required sample size (Taubenfeld et al., 2025b). In parallel, other approaches use an external verifier, like a PRM, to select the single highest-scoring candidate (Uesato et al., 2022). Our work builds on the idea of weighted voting, but instead of relying on the LLM's self-assessment, we derive weights from a principled theoretical framework that combines both the LLM's consensus signal and the external verifier's scores.

**Reward Modeling.** A crucial component of many TTS strategies is an external verifier, or reward model (RM), trained to score the quality of generated responses. An early, influential work by Cobbe et al. (2021) demonstrated that training a dedicated verifier to select the best solution from

many candidates could improve performance on math word problems more effectively than fine-tuning the generator itself Uesato et al. (2022). This spurred a distinction between two supervision strategies: Outcome Reward Models (ORMs), which are trained on the correctness of the final answer, and Process Reward Models (PRMs), which are trained on step-by-step human feedback. An initial comparison by Uesato et al. (2022) found that ORMs could achieve similar final-answer accuracy with less supervision, but PRMs were necessary to ensure the faithfulness of the reasoning process (Zheng et al., 2024). Subsequent work by Lightman et al. (2024) solidified the superiority of PRMs on more challenging tasks, establishing process supervision as a key technique for building reliable verifiers, despite its high annotation cost (Wang et al., 2024). Our work focuses on how to best leverage the signals from these powerful but expensive-to-train PRMs.

## 3 Optimal Response Aggregation

In this section, we aim to explore the optimal aggregation strategy for signals from the LLM and PRM. We start by formalizing this as a Maximum a Posteriori (MAP) estimation problem, and derive an optimal aggregation strategy in Section 3.1. Then, in Section 3.2, we manage to estimate the quantities in the optimal aggregation strategy empirically and offer a few critical insights on the optimal weighting strategy.

### 3.1 Theoretical Analysis of Optimal Response Aggregation

**Problem Setup.** Let $M$ be the LLM and $V$ be the PRM (Verifier). For a single prompt, $M$ generates an ensemble of $L$ responses, $\mathcal{G} = \{g_1, g_2, \ldots, g_L\}$. Each response $g_i$ consists of a reasoning process $r_i$ and a final answer $s_i = f(r_i)$. The PRM $V$ evaluates each generation $g_i$ and produces a scalar score $p_i$. Let $\mathcal{P} = \{p_1, p_2, \ldots, p_L\}$ be the set of these scores. The set of unique candidate answers is $\mathcal{A} = \{\alpha_1, \ldots, \alpha_m\}$. Our objective is to determine the most probable true answer $\hat{\alpha}$ given all available evidence.

We aim to find the answer $\alpha_k$ that maximizes the posterior probability $P(\alpha_k|\mathcal{G}, \mathcal{P}, M, V)$. By Bayes' theorem, and assuming a uniform prior over answers $P(\alpha_k|M, V)$, this is equivalent to maximizing the likelihood of the evidence:

$$\hat{\alpha} = \arg\max_{\alpha_k \in \mathcal{A}} P(\mathcal{G}, \mathcal{P}|\alpha_k, M, V) \tag{1}$$

We can decompose this likelihood into two factors: $P(\mathcal{G}, \mathcal{P}|\alpha_k, M, V) = P(\mathcal{P}|\mathcal{G}, \alpha_k, V) \times P(\mathcal{G}|\alpha_k, M)$. This reflects the causal process: the LLM $M$ generates responses $\mathcal{G}$, and then the Verifier $V$ produces scores $\mathcal{P}$ based on $\mathcal{G}$. To make this tractable, we introduce two conditional independence assumptions:

**Assumption 3.1** (Score and Generation Independence). *The PRM score $p_i$ for a generation $g_i$ is conditionally independent of all other generations, given $g_i$ and the true answer $\alpha_k$. The LLM generations $g_i$ are conditionally independent of each other, given the true answer $\alpha_k$.*

$$P(\mathcal{P}|\mathcal{G}, \alpha_k, V) = \prod_{i=1}^{L} P(p_i|g_i, \alpha_k, V), P(\mathcal{G}|\alpha_k, M) = \prod_{i=1}^{L} P(g_i|\alpha_k, M)$$

With these assumptions, the log-likelihood becomes a sum over individual responses: $\mathrm{LL}(\alpha_k) = \sum_{i=1}^{L} \log P(p_i|g_i, \alpha_k, V) + \sum_{i=1}^{L} \log P(g_i|\alpha_k, M)$. We hypothesize that under the condition $\alpha_k$, a generation $g_i$ with answer $s_i = \alpha_k$ is correct ($c_i = 1$), and incorrect ($c_i = 0$) otherwise. The term $P(g_i|\alpha_k, M)$ is simplified to $P(c_i|\alpha_k, M)$, where we assume a simple probability model for the LLM: it produces the correct answer with probability $q_M$ and any specific incorrect answer with probability $(1 - q_M)/(m - 1)$.

**Theorem 3.2** (Optimal Aggregation Score). *Under the assumptions above, maximizing the log-likelihood is equivalent to maximizing the score:*

$$Score(\alpha_k) = \sum_{i:s_i=\alpha_k} w_i, \quad where \ w_i = \underbrace{\log \frac{P(p_i|c_i=1, V)}{P(p_i|c_i=0, V)}}_{PRM \ Signal \ Term} + \underbrace{\log \frac{q_M \cdot (m-1)}{1 - q_M}}_{LLM \ Signal \ Term}$$

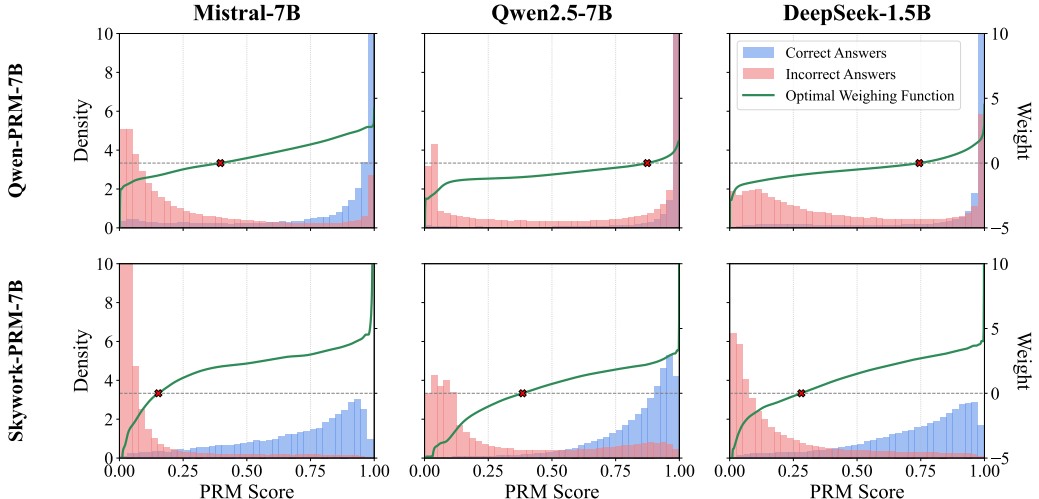

Figure 1: The PRM score distributions and optimal weighing functions on 6 combinations of LLM-PRM pairs. **Left y-axis:** the probability density of the PRM scores. **Right y-axis:** the optimal weights $w^*(p)$ learned via KDE for different LLM-PRM pairs. The zero-crossing point is marked with a red cross. Note their model-dependent nature and the presence of negative weights for low PRM scores.

*Proof.* The full derivation is in Appendix A. The key insight is that the log-likelihood can be rearranged into a sum of weights for responses voting for $\alpha_k$, plus terms that are constant with respect to $\alpha_k$ and can be dropped from the argmax. The weight $w_i$ for each vote combines signals from two sources: the PRM's score (via the likelihood ratio of the score $p_i$ occurring for correct vs. incorrect reasoning) and the LLM's intrinsic reliability (via the term involving $q_M$), also referred to as question difficulty (Snell et al., 2025). □

## 3.2 EMPIRICAL ANALYSIS OF THE OPTIMAL WEIGHTING FUNCTION

**Instantiating the optimal weights**. To instantiate the optimal weight $w_i$ from Theorem 3.2, we perform a *per-question estimation* using the ground-truth labels for the specific set of $L$ responses generated for that question. To estimate the PRM Signal term, we apply a separate Kernel Density Estimation (KDE) on the logit space for each individual question to estimate the score distributions $P(p|c, V)$ on the specific question. For details of our KDE estimation in the logit space, please refer to Section 4.1. To estimate the LLM Signal term, we simply set $q_M$ to be the true accuracy of the $L$ responses for that specific question.

**Characterizing the optimal weighting function**. Taking the PRM's ability to distinguish correct and wrong answers into account, this optimal aggregator provides a much tighter performance upper bound than Pass@N, demonstrating the potential of our framework, as shown in Figure 2. More importantly, analyzing the structure of the learned weighting function $w^*(p)$ (Figure 1) reveals two critical insights:

- **Weighting functions are highly model-dependent.** The shape of the optimal function varies dramatically depending on the specific LLM and PRM being used. A simple, fixed function (e.g., using the PRM score directly as a weight) is unlikely to be optimal across different model pairs. This underscores the necessity of a calibration procedure tailored to the specific models in use.
- **Presence of negative weights.** An interesting and consistent finding is that low PRM scores are mapped to negative weights. This implies that a response deemed incorrect by the PRM provides strong evidence *against* its proposed answer, and repetition of low-quality responses does not add to the likelihood of their answer being correct. Standard methods like Best-of-N, which only consider the top-scoring candidate, or majority voting, which ignores scores entirely, fail to

leverage this powerful negative signal. An effective aggregation strategy must penalize answers supported by low-quality reasoning.

These findings motivate the need for a practical method that can approximate these complex, non-linear, and often negative weighting functions without requiring ground-truth labels at test time. We address this challenge in the following section.

# 4 PRACTICAL CALIBRATION METHODS

The optimal analysis confirmed the need for a calibrated, model-specific weighting function. We now introduce practical methods to learn these functions using a one-time, pre-computed calibration set $D_{cal} = \{(r_1, p_1, c_1), ..., (r_n, p_n, c_n)\}$. Once a weighting function $w(p)$ is learned, the final answer is selected by a weighted vote: $\hat{\alpha} = \arg\max_{\alpha_k \in \mathcal{A}} \sum_{i:s_i = \alpha_k} w(p_i)$.

## 4.1 NON-PARAMETRIC WEIGHTING FUNCTIONS

One of the most straightforward ways towards the optimal aggregation strategy is to directly estimate the unknown quantities in the optimal weighting function 3.2, i.e., PRM score distributions $P(p|c = 1, V)$, $P(p|c = 0, V)$, and LLM reliability $q_M$.

**Estimating PRM score distribution.** To capture the nuances of the PRM score distribution on different LLM and PRM combinations, we apply the Kernel Density Estimation (KDE). Compared to other estimation methods, such as histogram estimation or parametric estimations, KDEs are smooth, continuous, and more flexible. However, while the PRM score is within the probability space between 0 and 1, KDEs are not bounded, spilling probability density outside this range. Consequently, we first convert the scores from the probability space to the logit space with the logit function $\text{logit}(p) = \log(\frac{p}{1-p})$. Then, we perform KDE of the scores within the logit space. Specifically, the PRM score distribution is estimated as:

$$\hat{f}_c(p) = \frac{1}{|D_c| \cdot h} \sum_{i \in D_c} K\left(\frac{\text{logit}(p) - \text{logit}(p_i)}{h}\right) \tag{2}$$

where $D_c = \{i | c_i = c\}$ separates responses within the calibration set $D_{cal}$ according to their correctness $c_i$. $K$ and $h$ are the kernel and bandwidth of KDE.

**Estimating LLM reliability.** To estimate $q_M$ at test time without labels, we first train a simple binned probability calibrator $g(\cdot)$ on the PRM scores from the calibration set. During inference, we calculate the calibrated probability for each of the $L$ generated responses $D_{test} = \{(r'_1, p'_1), ..., (r'_L, p'_L)\}$ to the test question and approximate $q_M$ as their average, i.e., $\hat{q}_M = \frac{1}{|D_{test}|} \sum_{i \in D_{test}} g(p'_i)$.

Given the estimations above, according to Equation 3.2, we have the estimated weighting function:

$$w_{KDE}(p) = \log(\hat{f}_1(p)) - \log(\hat{f}_0(p)) + \log(\hat{q}_M) + \log(m - 1) - \log(1 - \hat{q}_M)$$

This KDE is the practical counterpart to the optimal estimator, where the PRM score distribution is also estimated on $D_{test}$ with additional access to the correctness of test responses $c'_i$.

## 4.2 PARAMETRIC WEIGHTING FUNCTIONS

As an alternative, we explore simpler parametric forms for $w(p)$, optimizing parameters on the calibration set via grid search. These methods are guided by our insight about the importance of a zero-crossing point, controlled by the parameter $b$. This parameter acts as a threshold, making the weight positive for scores above it and negative for scores below, directly implementing the penalization of low-quality responses.

**Logit Weighting.** Inspired by the log-ratio form in our theorem, we model the weight as:

$$w_{logit}(p) = \text{logit}(p) - \text{logit}(b)$$

**Linear Weighting.** As a simpler baseline, we model the weight as:

$$w_{linear}(p) = p - b$$

During grid search, the parameter $b$ is searched within the range $[0, 1]$ and $[-1, 1]$ for Logit Weighted Voting (**Logit WV**) and Linear Weighted Voting (**Linear WV**), respectively. The objective used in the grid search is the accuracy on the held-out calibration set.

## 5 EXPERIMENTS

In this section, we first conduct a comprehensive evaluation of the scaling methods across 35 combinations of LLM and PRM in Section 5.2. Then we dive into the principles of the proposed methods in Section 5.3.

### 5.1 EXPERIMENTAL SETUP

**Models.** To capture the complexities of signal aggregation in practice, we use 5 LLMs across 3 model series (Mistral-7B (Wang et al., 2024), Qwen2.5-1.5B/7B (Yang et al., 2024), DeepSeek-1.5B/7B (DeepSeek-AI, 2025)) and 7 PRMs based on Qwen (Qwen2.5-PRM800K (Song et al., 2025), Qwen2.5-PRM-7B (Zhang et al., 2025b), Skywork-PRM-1.5/7B (He et al., 2024)) , Llama (Llama3.1-8B-PRM-Mistral/DeepSeek (Xiong et al., 2024)), and Mistral (math-shepherd-7b-prm (Wang et al., 2024)) series. During generation, we set the top-p and temperature configuration to 0.9 and 0.7, respectively.

**Data.** To simulate the scenarios where the reliability of the LLM signals varies, we evaluate performance on task with various difficulties, the MATH training set (MATH) and test set (MATH500) (Hendrycks et al., 2021). For the MATH dataset, we select the first 2k out of 7.5k questions as the pool for calibration, and randomly sample 1k questions from the pool as the calibration set for each run. Similarly, for MATH500, the calibration pool size is 200, and we randomly sample 100 questions from the pool as the calibration set. Questions outside of the pool are used as the test set. We run each experiment 3 times with different random seeds. For the MATH500 dataset, we sample 112 responses from each LLM for each question. For the MATH dataset, we sample 32 responses from each LLM for each question, due to its large size. Then, these collected responses are scored by each of the 7 PRMs. As such, we ensure all the scaling methods are using identical responses and PRM scores for fair comparison.

**Baselines.** We compare our calibrated weighted voting against several methods:

- **Majority Vote**: The answer with the most votes is selected, ignoring PRM scores. This is the standard self-consistency approach. $\hat{\alpha} = \arg\max_{\alpha_k \in \mathcal{A}} \sum_{i=1}^{L} \mathbb{I}(s_i = \alpha_k)$.
- **Best-of-N (BoN)**: The answer from the single response with the highest PRM score is chosen. $\hat{\alpha} = s_{i^*}$ where $i^* = \arg\max_i p_i$.
- **Vanilla Weighted Vote**: A weighted vote where the raw, uncalibrated PRM score $p_i$ is used as the weight. $\hat{\alpha} = \arg\max_{\alpha_k \in \mathcal{A}} \sum_{i:s_i = \alpha_k} p_i$.

We also report two theoretical bounds: **Pass@N**, which considers a problem solved if at least one response is correct, and **Optimal** as discussed in Section 3.2.

### 5.2 MAIN RESULTS

**Weighting function calibration significantly boosts TTS efficiency.** We evaluate the effectiveness of the proposed weighting function calibration methods by performing calibration for each LLM and PRM pair before scaling test-time compute. As shown in Figure 2, calibrating before scaling significantly boosts the efficiency. In particular, on the MATH and MATH500 datasets, the logit-based calibration method surpasses the performance of vanilla weighting voting methods *with approximately 37.1% and 21.3% of test-time compute on average across 35 LLM-PRM pairs*.

For detailed results on LLM-PRM pairs, we show the performance of various scaling methods in Table 1. We can see that calibration methods consistently outperform baseline scaling methods across LLMs and PRMs. Generally, Logit Weighted Voting performs the best in most cases. In particular, on the Llama3.1-Mistral-8B PRM, Logit WV outperforms the best-performing baseline method, Vanilla WV, by 3 points of accuracy (61.2 v.s. 58.2) on average across 5 LLMs, *which would otherwise take an exponential amount of test-time compute to achieve*. This strongly supports

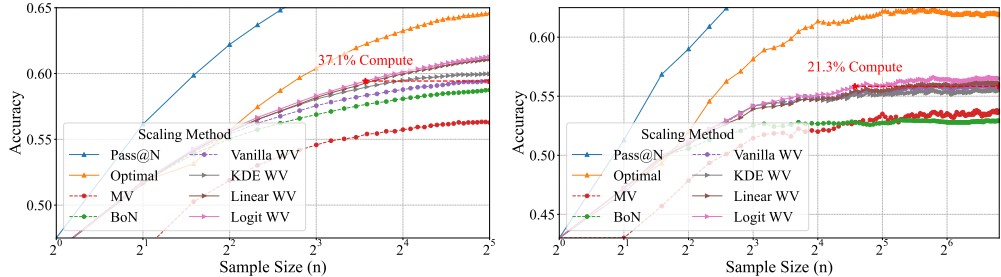

Figure 2: The performance of various scaling methods averaged across all LLM and PRM combinations. The computation efficiency improvement of the Logit WV compared to the best-performing baseline, Vanilla WV, is marked in red. **Left:** On the MATH dataset. **Right:** On the MATH500 dataset.

Table 1: Accuracy of TTS methods at sample size 32 on MATH. The best method for each case is in bold.

| PRM | Method | Mistral-7B | Qwen2.5-1.5B | Qwen2.5-7B | DeepSeek-1.5B | DeepSeek-7B | Average |
|---|---|---|---|---|---|---|---|
| Qwen- PRM800K- 7B | Optimal | 55.6 ± 0.0 | 64.5 ± 0.0 | 71.7 ± 0.0 | 59.3 ± 0.0 | 57.2 ± 0.0 | 61.7 ± 0.0 |
| | BoN | 49.5 ± 0.0 | 57.3 ± 0.0 | 65.3 ± 0.0 | 45.0 ± 0.0 | 46.1 ± 0.0 | 52.7 ± 0.0 |
| | MV | 51.3 ± 0.0 | 60.6 ± 0.0 | 67.8 ± 0.0 | **54.1 ± 0.0** | 51.9 ± 0.0 | 57.1 ± 0.0 |
| | Vanilla WV | 52.5 ± 0.0 | 61.4 ± 0.0 | 67.7 ± 0.0 | 53.7 ± 0.0 | 51.7 ± 0.0 | 57.4 ± 0.0 |
| | KDE WV | 52.3 ± 0.2 | 61.2 ± 0.1 | **68.0 ± 0.1** | 54.0 ± 0.1 | **52.0 ± 0.1** | 57.5 ± 0.0 |
| | Linear WV | 53.0 ± 0.0 | 61.5 ± 0.1 | 67.9 ± 0.0 | 53.8 ± 0.0 | 51.8 ± 0.0 | 57.6 ± 0.0 |
| | Logit WV | **53.0 ± 0.0** | **61.5 ± 0.0** | 67.7 ± 0.1 | 53.9 ± 0.1 | 51.9 ± 0.0 | **57.6 ± 0.0** |
| Qwen- PRM- 7B | Optimal | 58.5 ± 0.0 | 67.9 ± 0.0 | 75.4 ± 0.0 | 67.6 ± 0.0 | 63.1 ± 0.0 | 66.5 ± 0.0 |
| | BoN | 53.7 ± 0.0 | 62.6 ± 0.0 | 70.4 ± 0.0 | 62.5 ± 0.0 | 59.8 ± 0.0 | 61.8 ± 0.0 |
| | MV | 51.3 ± 0.0 | 60.6 ± 0.0 | 67.8 ± 0.0 | 54.1 ± 0.0 | 51.9 ± 0.0 | 57.1 ± 0.0 |
| | Vanilla WV | 54.3 ± 0.0 | 62.9 ± 0.0 | 70.2 ± 0.0 | 59.9 ± 0.0 | 56.8 ± 0.0 | 60.8 ± 0.0 |
| | KDE WV | 52.8 ± 0.1 | 61.9 ± 0.1 | 68.8 ± 0.1 | 59.1 ± 0.4 | 58.2 ± 1.2 | 60.2 ± 0.2 |
| | Linear WV | 54.1 ± 0.0 | 63.0 ± 0.1 | 70.3 ± 0.0 | 64.7 ± 0.1 | 62.6 ± 0.0 | 62.9 ± 0.0 |
| | Logit WV | **54.3 ± 0.0** | **63.5 ± 0.1** | **70.5 ± 0.2** | **65.1 ± 0.0** | **62.9 ± 0.0** | **63.3 ± 0.0** |
| Llama3.1- Mistral- 8B | Optimal | 58.7 ± 0.0 | 65.8 ± 0.0 | 74.4 ± 0.0 | 65.7 ± 0.0 | 61.9 ± 0.0 | 65.3 ± 0.0 |
| | BoN | 54.2 ± 0.0 | 58.0 ± 0.0 | 68.7 ± 0.0 | 56.9 ± 0.0 | 53.5 ± 0.0 | 58.3 ± 0.0 |
| | MV | 51.3 ± 0.0 | 60.6 ± 0.0 | 67.8 ± 0.0 | 54.1 ± 0.0 | 51.9 ± 0.0 | 57.1 ± 0.0 |
| | Vanilla WV | 53.9 ± 0.0 | 62.2 ± 0.0 | 69.4 ± 0.0 | 56.9 ± 0.0 | 54.8 ± 0.0 | 59.4 ± 0.0 |
| | KDE WV | 53.9 ± 0.3 | 62.0 ± 0.1 | 68.8 ± 0.1 | 58.0 ± 0.1 | 58.1 ± 0.6 | 60.2 ± 0.2 |
| | Linear WV | 54.7 ± 0.0 | 62.2 ± 0.5 | 70.1 ± 0.4 | 62.3 ± 0.0 | 60.2 ± 0.0 | 61.9 ± 0.2 |
| | Logit WV | **55.6 ± 0.1** | **62.9 ± 0.0** | **70.7 ± 0.3** | **62.8 ± 0.1** | **60.7 ± 0.1** | **62.6 ± 0.1** |
| Llama3.1- DS- 8B | Optimal | 58.1 ± 0.0 | 65.9 ± 0.0 | 73.8 ± 0.0 | 65.6 ± 0.0 | 61.6 ± 0.0 | 65.0 ± 0.0 |
| | BoN | 51.3 ± 0.0 | 59.5 ± 0.0 | 68.8 ± 0.0 | 59.0 ± 0.0 | 55.9 ± 0.0 | 58.9 ± 0.0 |
| | MV | 51.3 ± 0.0 | 60.6 ± 0.0 | 67.8 ± 0.0 | 54.1 ± 0.0 | 51.9 ± 0.0 | 57.1 ± 0.0 |
| | Vanilla WV | 54.3 ± 0.0 | **61.7 ± 0.0** | 69.6 ± 0.0 | 57.1 ± 0.0 | 55.7 ± 0.0 | 59.7 ± 0.0 |
| | KDE WV | 53.3 ± 0.6 | 61.4 ± 0.1 | 68.7 ± 0.0 | 58.3 ± 0.2 | 56.9 ± 0.4 | 59.7 ± 0.2 |
| | Linear WV | 54.7 ± 0.1 | 61.5 ± 0.2 | 69.9 ± 0.1 | 60.7 ± 0.6 | 58.9 ± 0.1 | 61.1 ± 0.1 |
| | Logit WV | **54.9 ± 0.0** | 61.6 ± 0.0 | **69.9 ± 0.0** | **61.9 ± 0.1** | **59.2 ± 0.3** | **61.5 ± 0.0** |
| Skywork- PRM- 1.5B | Optimal | 57.7 ± 0.0 | 73.1 ± 0.0 | 75.1 ± 0.0 | 67.3 ± 0.0 | 68.9 ± 0.0 | 68.4 ± 0.0 |
| | BoN | 54.2 ± 0.0 | 70.3 ± 0.0 | **72.5 ± 0.0** | 63.5 ± 0.0 | **66.1 ± 0.0** | 65.3 ± 0.0 |
| | MV | 51.3 ± 0.0 | 65.8 ± 0.0 | 67.8 ± 0.0 | 54.1 ± 0.0 | 57.1 ± 0.0 | 59.2 ± 0.0 |
| | Vanilla WV | 54.9 ± 0.0 | 69.4 ± 0.0 | 71.6 ± 0.0 | 60.3 ± 0.0 | 61.5 ± 0.0 | 63.5 ± 0.0 |
| | KDE WV | 53.8 ± 0.1 | 68.4 ± 0.4 | 70.6 ± 0.1 | 60.6 ± 1.0 | 62.8 ± 1.3 | 63.2 ± 0.5 |
| | Linear WV | **55.8 ± 0.0** | **70.3 ± 0.1** | 72.2 ± 0.3 | **66.2 ± 0.1** | 65.5 ± 0.1 | **66.0 ± 0.1** |
| | Logit WV | 55.6 ± 0.0 | 70.2 ± 0.0 | 72.0 ± 0.2 | 66.0 ± 0.0 | 65.5 ± 0.1 | 65.9 ± 0.0 |

our claim of calibrating the weighting function before expensive TTS. Please refer to Appendix B for more detailed results.

**Consistently effective on domains other than math.** To demonstrate broader applicability, we further experimented on a wide range of tasks beyond math. In the MMLU-Pro-CoT-Eval dataset, there are approximately 150 questions in each task, among which 50 are used as the calibration set and the rest as the test set. Llama-3.1-8B-Instruct and VersaPRM are the LLM-PRM pair. As seen in Table 2, the proposed methods are consistently effective on a wide range of tasks beyond the math domain.

Table 2: Accuracy of aggregation methods at sample size n=16 for Llama-3.1-8B-Instruct and VersaPRM on the MMLU-Pro-CoT-Eval dataset.

| PRM | Method | Health | CS | Econ | Chem | Other | Physics | Law | Eng | History | Math | Phil | Average |
|---|---|---|---|---|---|---|---|---|---|---|---|---|---|
| VersaPRM | Optimal | 71.8 | 78.0 | 76.0 | 85.0 | 79.0 | 84.0 | 65.3 | 78.0 | 62.0 | 85.0 | 70.7 | 75.9 |
| | BoN | 61.2 | 57.0 | 62.0 | 64.0 | 61.0 | 62.0 | 45.3 | 55.0 | 46.0 | 63.0 | 50.5 | 57.0 |
| | MV | 62.4 | 55.0 | 61.0 | 64.0 | 57.0 | 62.0 | 35.8 | 56.0 | 44.0 | 61.0 | 42.4 | 54.6 |
| | Vanilla WV | 63.5 | 57.0 | 62.0 | 66.0 | 59.0 | 64.0 | 43.2 | 60.0 | 46.0 | 65.0 | 48.5 | 57.7 |
| | KDE WV | **64.7** | 57.0 | 63.0 | **66.0** | 58.0 | 64.0 | 42.1 | 58.0 | 45.0 | 65.0 | 46.5 | 57.2 |
| | Linear WV | 63.5 | 57.0 | **64.0** | 66.0 | **62.0** | **67.0** | 44.2 | **63.0** | **47.0** | **66.0** | 46.5 | 58.7 |
| | Logit WV | 62.4 | **58.0** | 64.0 | 65.0 | 60.0 | 66.0 | **45.3** | 63.0 | 44.0 | 66.0 | **53.5** | **58.8** |

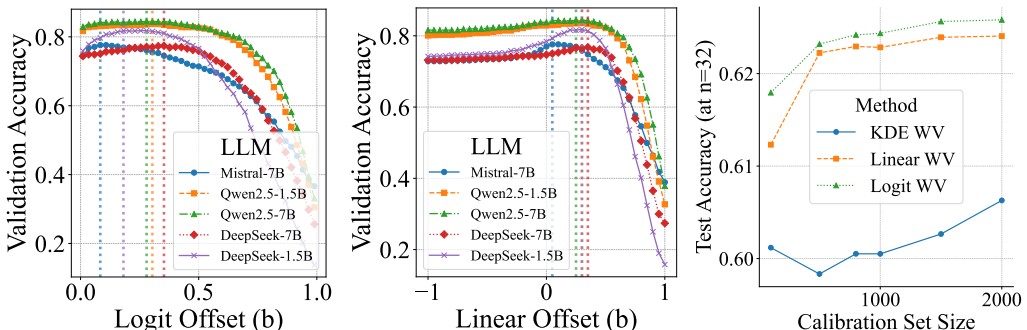

Figure 3: **Left and Middle:** The grid search result of the offset parameter $b$ for both Logit WV and Linear WV, where the optimal value is marked with vertical lines. The consistently positive optimal value across LLMs demonstrates the necessity of negative weights. **Right:** The performance of the calibration methods when we scale the calibration set size. The performance can be further improved with larger calibration sets.

## 5.3 EMPIRICAL ANALYSIS

**Are negative weights necessary in utilizing the PRM signals?** To further verify our insight that negative weights are necessary for better utilization of the PRM signals, we show the grid search result on the offset parameter $b$ of Logit WV and Linear WV, where, in both cases, its value suggests the zero-crossing point, assigning negative weights to responses whose PRM score is lower than $b$. As shown in Figure 3, for both weighting functions, the optimal offset $b$ is consistently larger than zero across all LLMs, proving the necessity of negative weights in efficient TTS. Furthermore, the zero-crossing points are different for each LLM, but are generally consistent for the same LLM using different weighting functions (Linear and Logit), demonstrating the PRM's varying capability in distinguishing positive and negative responses from different LLMs. This further supports our claim to take the unique interactions between the models into account, i.e., calibration, for efficient TTS.

**What's the remaining gap and challenges towards the optimal weighting function?** To answer this question, we compare our dataset-wise estimation of the PRM score weighting function $\log\left(\frac{f_1(p)}{f_0(p)}\right)$ with the optimal per-question estimation on several questions. As shown in the left subplot of Figure 4, while our estimation captures the dataset-wise PRM score weighting function, the optimal weighting function for each individual question varies largely, suggesting a global scoring function is still suboptimal. We also examine how accurate our estimation of the LLM reliability term $q_M$ is with Mean Absolute Error. As shown in the right subplot of Figure 4, while compared to a fixed global LLM reliability, using calibrated PRM scores to estimate this term effectively reduces the error, the error for most PRMs is still larger than 0.2, which is far from negligible. This also explains the relatively low performance of KDE WV compared to the parametric counterparts. In conclusion, accurately estimating either the PRM or the LLM part of the weight requires nuanced estimation for individual questions, explaining why the non-parametric KDE estimation underperforms the parametric ones. We find such per-question estimation inherently difficult in our attempts

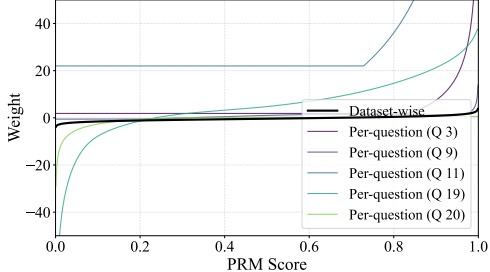 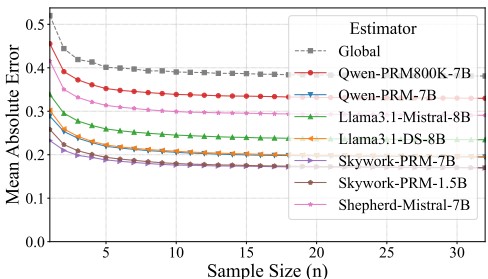

Figure 4: **Left:** The comparison of the dataset-wise estimated PRM score weighting function and the per-question estimated optimal PRM score weighting function. A large variance among questions can be seen. **Right:** The mean absolute error of estimated $\hat{q}_M$ compared to the true $q_M$.

Table 3: Average accuracy of out-of-domain test tasks of Aggregation Methods at n=16 for LLM 'Llama-3.1-8B-Instruct'

| Method | Chem | CS | Econ | Eng | Health | History | Law | Math | Other | Phil | Physics | Average |
|---|---|---|---|---|---|---|---|---|---|---|---|---|
| Optimal | 74.8 | 75.2 | 75.5 | 75.3 | 75.7 | 76.5 | 76.8 | 74.8 | 75.3 | 76.0 | 74.8 | 75.5 |
| BoN | 56.4 | 56.8 | 56.5 | 57.2 | 56.8 | 57.7 | 58.2 | 56.5 | 56.6 | 58.0 | 56.4 | 57.0 |
| MV | 53.7 | 54.3 | 53.9 | 54.3 | 53.3 | 55.1 | 56.3 | 53.8 | 54.2 | 55.5 | 53.5 | 54.4 |
| Vanilla WV | 57.0 | 57.5 | 57.3 | 57.5 | 56.8 | 58.4 | 59.1 | 57.0 | 57.5 | 58.7 | 56.8 | 57.6 |
| KDE WV | 57.5 | 56.3 | 55.0 | 57.7 | 55.0 | 57.5 | 59.8 | 56.9 | 56.3 | 57.8 | 57.4 | 57.0 |
| Linear WV | **57.7** | **57.9** | **58.1** | 57.7 | 55.2 | **59.5** | 58.9 | 57.1 | **58.0** | 58.4 | 56.7 | 57.8 |
| Logit WV | 57.7 | 57.8 | 57.9 | **58.1** | **57.8** | 59.4 | **59.9** | 57.5 | 57.9 | **58.8** | 57.7 | **58.2** |
| **Average** | 56.7 | 56.7 | 56.5 | 57.1 | 55.8 | 57.9 | 58.7 | 56.5 | 56.7 | 57.9 | 56.4 | 57.0 |

to learn a meta model to predict per-question weighting functions, which struggles to fit and generalize.

**How does calibration set size affect the performance?** We rearrange the split of the MATH dataset to reserve 5k questions as the test, and the rest as the pool of calibration data. As shown in the right subplot of Figure 3, while the validation set used in the main experiments is relative small, the performances of the calibration methods can be further enhanced if we scale the calibration set size to calibrate the weighting function better.

**Are calibration results transferable to out-of-domain test sets?** To further examine the effectiveness of calibration sets that are not in the same domain as the test set, we calibrate an aggregator on each of the tasks, and report its average accuracy when tested on the other tasks in the following table, where each column corresponds to the calibrator trained on that task. As we can see in Table 3, despite being calibrated on various "Out-of-domain" calibration sets, the proposed methods consistently outperform the baselines, showing strong transferability.

## 6 CONCLUSION

We address the suboptimal use of Process Reward Models (PRMs) in Test-Time Scaling (TTS). Through a theoretical MAP framework, we show that the optimal aggregation strategy is a weighted majority vote combining signals from both the LLM and PRM. Empirically, we find these optimal weights are model-dependent and, critically, assign large negative values to penalize low-quality responses—a powerful signal neglected by standard methods. We propose simple calibration methods to learn these functions. Our calibrated weighted voting boosts TTS efficiency, achieving superior accuracy over baselines like Best-of-N with approximately 37.1% and 21.3% computational cost. This work demonstrates that intelligent aggregation is a more efficient path to performance gains than simply scaling test-time compute.

## ACKNOWLEDGMENT

This work was partially supported by the National Artificial Intelligence Research Resource (NAIRR) Pilot under awards NAIRR250400 and NAIRR240283, and Standing Up to POTS.

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

# A PROOFS

## A.1 DERIVATION OF THEOREM 3.2

Our objective is to find the answer $\hat{\alpha}$ that maximizes the posterior probability $P(\alpha_k|\mathcal{G}, \mathcal{P}, M, V)$. Assuming a uniform prior over answers, this is equivalent to maximizing the log-likelihood, $\text{LL}(\alpha_k)$. From the main text, the log-likelihood under Assumption 3.1 is:

$$\text{LL}(\alpha_k) = \sum_{i=1}^{L} \log P(p_i|g_i, \alpha_k, V) + \sum_{i=1}^{L} \log P(g_i|\alpha_k, M) \tag{3}$$

Let $c_i \in \{0, 1\}$ be a binary variable indicating whether generation $g_i$ is correct ($c_i = 1$) or incorrect ($c_i = 0$). Under the hypothesis that the true answer is $\alpha_k$, the correctness of $g_i$ is determined by its answer $s_i$. Specifically, $c_i = 1$ if $s_i = \alpha_k$, and $c_i = 0$ if $s_i \neq \alpha_k$.

We can now analyze the two components of the log-likelihood separately.

**Part 1: The PRM Signal Term**

The first sum can be split based on whether a generation's answer $s_i$ matches the candidate answer $\alpha_k$:

$$\sum_{i=1}^{L} \log P(p_i|g_i, \alpha_k, V) = \sum_{i:s_i=\alpha_k} \log P(p_i|g_i, c_i = 1, V) + \sum_{i:s_i\neq\alpha_k} \log P(p_i|g_i, c_i = 0, V) \tag{4}$$

To isolate the terms relevant to the maximization over $\alpha_k$, we rewrite the second sum by noting that $\sum_{i:s_i\neq\alpha_k}(\cdot) = \sum_{i=1}^{L}(\cdot) - \sum_{i:s_i=\alpha_k}(\cdot)$:

$$= \sum_{i:s_i=\alpha_k} \log P(p_i|g_i, c_i = 1, V) + \sum_{i=1}^{L} \log P(p_i|g_i, c_i = 0, V) - \sum_{i:s_i=\alpha_k} \log P(p_i|g_i, c_i = 0, V)$$

$$= \sum_{i:s_i=\alpha_k} (\log P(p_i|g_i, c_i = 1, V) - \log P(p_i|g_i, c_i = 0, V)) + \sum_{i=1}^{L} \log P(p_i|g_i, c_i = 0, V) \tag{5}$$

The second term, $\sum_{i=1}^{L} \log P(p_i|g_i, c_i = 0, V)$, is a sum over all $L$ generations. Since this term does not depend on the choice of the candidate answer $\alpha_k$, it is a constant with respect to our maximization problem and can be dropped. This leaves us with the $\alpha_k$-dependent part of the PRM signal:

$$\text{PRM\_Term}(\alpha_k) = \sum_{i:s_i=\alpha_k} \log \frac{P(p_i|g_i, c_i = 1, V)}{P(p_i|g_i, c_i = 0, V)} \tag{6}$$

**Part 2: The LLM Signal Term**

Next, we analyze the LLM term, simplifying $P(g_i|\alpha_k, M)$ to $P(s_i|\alpha_k, M)$. We use the model's probabilities: $P(s_i = \alpha_k|\alpha_k, M) = q_M$ and, for any $s_j \neq \alpha_k$, $P(s_j|\alpha_k, M) = (1 - q_M)/(m - 1)$. Let $N_k$ be the count of generations where the answer is $\alpha_k$, i.e., $N_k = |\{i|s_i = \alpha_k\}|$.

$$\sum_{i=1}^{L} \log P(s_i|\alpha_k, M) = \sum_{i:s_i=\alpha_k} \log P(s_i = \alpha_k|\alpha_k, M) + \sum_{i:s_i\neq\alpha_k} \log P(s_i \neq \alpha_k|\alpha_k, M)$$

$$= N_k \log q_M + (L - N_k) \log \frac{1 - q_M}{m - 1}$$

$$= N_k \log q_M + L \log \frac{1 - q_M}{m - 1} - N_k \log \frac{1 - q_M}{m - 1}$$

$$= N_k \left( \log q_M - \log \frac{1 - q_M}{m - 1} \right) + L \log \frac{1 - q_M}{m - 1} \tag{7}$$

Similar to the PRM term, the second part, $L \log \frac{1-q_M}{m-1}$, does not depend on the specific candidate answer $\alpha_k$ (as $q_M, L, m$ are fixed for a given question) and can be dropped from the objective

function. The remaining term is:

$$\text{LLM\_Term}(\alpha_k) = N_k \log \frac{q_M \cdot (m-1)}{1 - q_M} = \sum_{i:s_i=\alpha_k} \log \frac{q_M \cdot (m-1)}{1 - q_M} \tag{8}$$

**Part 3: Combining the Terms**

Maximizing $\text{LL}(\alpha_k)$ is equivalent to maximizing the sum of the $\alpha_k$-dependent terms we derived. Let this new objective function be $\text{Score}(\alpha_k)$:

$$\begin{aligned}
\text{Score}(\alpha_k) &= \text{PRM\_Term}(\alpha_k) + \text{LLM\_Term}(\alpha_k) \\
&= \sum_{i:s_i=\alpha_k} \log \frac{P(p_i|c_i=1,V)}{P(p_i|c_i=0,V)} + \sum_{i:s_i=\alpha_k} \log \frac{q_M \cdot (m-1)}{1-q_M} \\
&= \sum_{i:s_i=\alpha_k} \left( \log \frac{P(p_i|c_i=1,V)}{P(p_i|c_i=0,V)} + \log \frac{q_M \cdot (m-1)}{1-q_M} \right)
\end{aligned} \tag{9}$$

This is a weighted majority vote, where the final score for an answer $\alpha_k$ is the sum of weights $w_i$ for all generations $g_i$ that produced that answer. The weight for each generation is:

$$w_i = \underbrace{\log \frac{P(p_i|c_i=1,V)}{P(p_i|c_i=0,V)}}_{\text{PRM Signal Term}} + \underbrace{\log \frac{q_M \cdot (m-1)}{1-q_M}}_{\text{LLM Signal Term}}$$

This completes the proof.

## B  ADDITIONAL EXPERIMENT RESULTS

### B.1  DETAILED RESULTS ON THE MATH500 DATASET

We show the detailed results on each LLM-PRM pair on the MATH500 dataset in Table 5.

### B.2  ANALYSIS ON KERNEL ESTIMATION BASED METHOD.

To validate the theoretical model itself and understand the relative underperformance of KDE WV, we ablate the optimal estimation with estimation from KDE. We add two baselines, KDE (Opt LLM term) and KDE (Opt PRM term), which substitute the LLM and PRM terms of KDE WV with optimal estimations, respectively. As shown in Table 4, using optimal estimation for the LLM and PRM term both lead to significantly better results, demonstrating that the gap between KDE and "Optimal" is indeed due to the difficulty of practical estimation, not a flaw in the theory.

### B.3  ADDITIONAL RESULT ON SELF-REWARDED VERIFICATION SIGNALS

CISC Taubenfeld et al. (2025a) is an approach that uses the confidence of the LLM as verification signals. While the verification signal is not from a PRM, our aggregation method is still applicable to CISC to utilize the signal more effectively. Here, we use the best-performing "P(True)" implementation of CISC to examine the effectiveness of the proposed method in aggregating self-rewarded verification signals. As we can see from the results on the MATH500 dataset in Table 6, the proposed method can more effectively aggregate the self-rewarded verification signal than vanilla weighted majority voting and Best-of-N.

### B.4  VARIANCE BETWEEN THE OPTIMAL WEIGHTING FUNCTION OF INDIVIDUAL QUESTIONS

In addition to the five qualitative examples, we have added a quantitative metric. We first scale the output range of the scoring functions to (-0.5, 0.5), and then compute the Mean Squared Error (MSE) between the dataset-wise estimated PRM score function and the per-question-optimal function, aggregated over all questions in the test set. This provides a more comprehensive view of the variance. As we can see in Table 7, the MSE between the dataset-wise estimated PRM score function and the per-question-optimal function is inherently significant, which validates our hypothesis that the optimal weighting function for each individual question varies largely, acting as one major challenge towards the optimal weighting function.

Table 4: Ablation of the Optimal estimation at Sample Size n=32

| PRM | Method | Mistral-7B | Qwen2.5-1.5B | Qwen2.5-7B | DeepSeek-1.5B | DeepSeek-7B | Average |
|---|---|---|---|---|---|---|---|
| Qwen- PRM800K- 7B | Optimal | 53.9 | 64.2 | 70.5 | 57.4 | 57.1 | 60.6 |
| | KDE WV (Opt PRM term) | 51.2 | 61.5 | 67.6 | 53.5 | 52.8 | 57.3 |
| | KDE WV (Opt LLM term) | 52.7 | 61.1 | 67.5 | 51.7 | 54.2 | 57.5 |
| | KDE WV | 50.1 | 60.6 | 66.8 | 51.8 | 51.6 | 56.2 |
| Qwen- PRM- 7B | Optimal | 56.8 | 67.7 | 74.5 | 66.4 | 63.6 | 65.8 |
| | KDE WV (Opt PRM term) | 54.0 | 65.2 | 71.4 | 61.3 | 61.2 | 62.6 |
| | KDE WV (Opt LLM term) | 53.7 | 63.1 | 69.4 | 63.2 | 62.8 | 62.4 |
| | KDE WV | 50.7 | 61.6 | 68.6 | 55.1 | 56.3 | 58.5 |
| Llama3.1- Mistral- 8B | Optimal | 57.1 | 65.6 | 73.4 | 64.3 | 62.2 | 64.5 |
| | KDE WV (Opt PRM term) | 55.1 | 63.0 | 70.6 | 58.5 | 58.5 | 61.2 |
| | KDE WV (Opt LLM term) | 54.5 | 62.8 | 69.0 | 62.0 | 62.3 | 62.1 |
| | KDE WV | 51.6 | 61.6 | 67.5 | 55.1 | 57.1 | 58.6 |
| Llama3.1- DS- 8B | Optimal | 56.4 | 65.7 | 73.0 | 64.2 | 62.0 | 64.2 |
| | KDE WV (Opt PRM term) | 54.1 | 62.9 | 70.0 | 58.4 | 58.5 | 60.8 |
| | KDE WV (Opt LLM term) | 53.8 | 61.8 | 69.1 | 61.2 | 60.6 | 61.3 |
| | KDE WV | 50.3 | 60.7 | 67.5 | 54.0 | 53.8 | 57.3 |
| Skywork- PRM- 1.5B | Optimal | 55.9 | 72.0 | 74.1 | 66.0 | 68.3 | 67.3 |
| | KDE WV (Opt PRM term) | 53.1 | 69.3 | 71.3 | 60.5 | 63.8 | 63.6 |
| | KDE WV (Opt LLM term) | 55.0 | 69.2 | 71.8 | 65.0 | 67.4 | 65.7 |
| | KDE WV | 51.7 | 66.3 | 69.8 | 57.3 | 59.3 | 60.9 |

Table 5: Accuracy of Aggregation Methods at Sample Size n=112, MATH500 dataset

| PRM | Method | Mistral-7B | Qwen2.5-1.5B | Qwen2.5-7B | DeepSeek-1.5B | DeepSeek-7B | Average |
|---|---|---|---|---|---|---|---|
| Qwen- PRM800K- 7B | Optimal | $37.3 \pm 0.0$ | $68.0 \pm 0.0$ | $69.7 \pm 0.0$ | $58.3 \pm 0.0$ | $62.7 \pm 0.0$ | $59.2 \pm 0.0$ |
| | BoN | $33.0 \pm 0.0$ | $57.0 \pm 0.0$ | $62.0 \pm 0.0$ | $40.3 \pm 0.0$ | $49.0 \pm 0.0$ | $48.3 \pm 0.0$ |
| | MV | $29.3 \pm 0.0$ | $61.3 \pm 0.0$ | $66.0 \pm 0.0$ | $52.7 \pm 0.0$ | $57.0 \pm 0.0$ | $53.3 \pm 0.0$ |
| | Vanilla WV | $31.7 \pm 0.0$ | $\mathbf{63.7 \pm 0.0}$ | $65.7 \pm 0.0$ | $53.0 \pm 0.0$ | $56.7 \pm 0.0$ | $54.1 \pm 0.0$ |
| | KDE WV | $31.3 \pm 1.5$ | $62.0 \pm 0.0$ | $\mathbf{66.3 \pm 0.0}$ | $\mathbf{53.0 \pm 0.0}$ | $57.0 \pm 0.0$ | $53.9 \pm 0.3$ |
| | Linear WV | $\mathbf{33.3 \pm 0.6}$ | $62.9 \pm 0.8$ | $66.3 \pm 0.0$ | $51.0 \pm 2.9$ | $\mathbf{57.0 \pm 0.0}$ | $\mathbf{54.1 \pm 0.5}$ |
| | Logit WV | $32.0 \pm 0.0$ | $62.9 \pm 0.4$ | $66.3 \pm 0.0$ | $52.7 \pm 0.0$ | $56.0 \pm 1.2$ | $54.0 \pm 0.3$ |
| Qwen- PRM- 7B | Optimal | $44.3 \pm 0.0$ | $70.3 \pm 0.0$ | $73.0 \pm 0.0$ | $67.7 \pm 0.0$ | $67.7 \pm 0.0$ | $64.6 \pm 0.0$ |
| | BoN | $\mathbf{40.0 \pm 0.0}$ | $63.7 \pm 0.0$ | $67.3 \pm 0.0$ | $59.7 \pm 0.0$ | $65.0 \pm 0.0$ | $59.1 \pm 0.0$ |
| | MV | $29.3 \pm 0.0$ | $61.3 \pm 0.0$ | $66.0 \pm 0.0$ | $52.7 \pm 0.0$ | $57.0 \pm 0.0$ | $53.3 \pm 0.0$ |
| | Vanilla WV | $35.3 \pm 0.0$ | $64.0 \pm 0.0$ | $67.3 \pm 0.0$ | $57.7 \pm 0.0$ | $62.3 \pm 0.0$ | $57.3 \pm 0.0$ |
| | KDE WV | $33.0 \pm 1.2$ | $63.0 \pm 0.9$ | $66.8 \pm 0.8$ | $56.1 \pm 0.2$ | $62.1 \pm 0.4$ | $56.2 \pm 0.4$ |
| | Linear WV | $36.0 \pm 0.0$ | $63.9 \pm 0.8$ | $67.8 \pm 0.4$ | $63.0 \pm 0.6$ | $\mathbf{65.7 \pm 0.0}$ | $59.3 \pm 0.2$ |
| | Logit WV | $37.6 \pm 0.4$ | $\mathbf{64.2 \pm 0.4}$ | $\mathbf{68.0 \pm 0.6}$ | $\mathbf{63.3 \pm 0.0}$ | $65.3 \pm 0.0$ | $\mathbf{59.7 \pm 0.2}$ |
| Llama3.1- Mistral- 8B | Optimal | $44.0 \pm 0.0$ | $65.3 \pm 0.0$ | $69.3 \pm 0.0$ | $63.0 \pm 0.0$ | $66.0 \pm 0.0$ | $61.5 \pm 0.0$ |
| | BoN | $\mathbf{32.7 \pm 0.0}$ | $50.0 \pm 0.0$ | $59.7 \pm 0.0$ | $49.3 \pm 0.0$ | $55.3 \pm 0.0$ | $49.4 \pm 0.0$ |
| | MV | $29.3 \pm 0.0$ | $\mathbf{61.3 \pm 0.0}$ | $66.0 \pm 0.0$ | $52.7 \pm 0.0$ | $57.0 \pm 0.0$ | $53.3 \pm 0.0$ |
| | Vanilla WV | $30.0 \pm 0.0$ | $60.0 \pm 0.0$ | $\mathbf{67.0 \pm 0.0}$ | $53.7 \pm 0.0$ | $58.7 \pm 0.0$ | $53.9 \pm 0.0$ |
| | KDE WV | $30.8 \pm 1.3$ | $61.0 \pm 0.0$ | $66.0 \pm 0.0$ | $54.0 \pm 0.0$ | $58.2 \pm 0.4$ | $54.0 \pm 0.3$ |
| | Linear WV | $31.0 \pm 0.3$ | $60.2 \pm 0.4$ | $66.0 \pm 0.0$ | $\mathbf{55.4 \pm 1.6}$ | $59.3 \pm 0.6$ | $54.4 \pm 0.4$ |
| | Logit WV | $31.8 \pm 0.4$ | $59.4 \pm 1.3$ | $66.3 \pm 0.6$ | $55.3 \pm 1.5$ | $\mathbf{59.4 \pm 0.4}$ | $\mathbf{54.5 \pm 0.3}$ |
| Llama3.1- DS- 8B | Optimal | $39.0 \pm 0.0$ | $67.7 \pm 0.0$ | $69.7 \pm 0.0$ | $61.3 \pm 0.0$ | $66.0 \pm 0.0$ | $60.7 \pm 0.0$ |
| | BoN | $27.0 \pm 0.0$ | $51.0 \pm 0.0$ | $62.3 \pm 0.0$ | $51.3 \pm 0.0$ | $59.0 \pm 0.0$ | $50.1 \pm 0.0$ |
| | MV | $29.3 \pm 0.0$ | $\mathbf{61.3 \pm 0.0}$ | $66.0 \pm 0.0$ | $52.7 \pm 0.0$ | $57.0 \pm 0.0$ | $53.3 \pm 0.0$ |
| | Vanilla WV | $\mathbf{30.0 \pm 0.0}$ | $59.3 \pm 0.0$ | $66.3 \pm 0.0$ | $55.3 \pm 0.0$ | $60.0 \pm 0.0$ | $54.2 \pm 0.0$ |
| | KDE WV | $29.8 \pm 0.2$ | $61.0 \pm 0.0$ | $66.2 \pm 0.2$ | $55.1 \pm 0.2$ | $58.9 \pm 1.1$ | $54.2 \pm 0.2$ |
| | Linear WV | $29.3 \pm 0.0$ | $61.0 \pm 0.0$ | $66.1 \pm 0.2$ | $55.0 \pm 1.2$ | $\mathbf{61.2 \pm 0.7}$ | $54.5 \pm 0.3$ |
| | Logit WV | $29.8 \pm 0.4$ | $60.1 \pm 0.4$ | $\mathbf{66.3 \pm 0.0}$ | $\mathbf{57.1 \pm 1.0}$ | $61.1 \pm 0.4$ | $\mathbf{54.9 \pm 0.2}$ |
| Skywork- PRM- 1.5B | Optimal | $40.7 \pm 0.0$ | $66.3 \pm 0.0$ | $71.0 \pm 0.0$ | $63.3 \pm 0.0$ | $66.3 \pm 0.0$ | $61.5 \pm 0.0$ |
| | BoN | $38.3 \pm 0.0$ | $56.7 \pm 0.0$ | $63.3 \pm 0.0$ | $54.0 \pm 0.0$ | $58.7 \pm 0.0$ | $54.2 \pm 0.0$ |
| | MV | $29.3 \pm 0.0$ | $61.3 \pm 0.0$ | $66.0 \pm 0.0$ | $52.7 \pm 0.0$ | $57.0 \pm 0.0$ | $53.3 \pm 0.0$ |
| | Vanilla WV | $35.7 \pm 0.0$ | $61.7 \pm 0.0$ | $\mathbf{68.3 \pm 0.0}$ | $58.0 \pm 0.0$ | $\mathbf{60.3 \pm 0.0}$ | $56.8 \pm 0.0$ |
| | KDE WV | $33.6 \pm 0.7$ | $61.3 \pm 0.0$ | $66.0 \pm 0.6$ | $55.6 \pm 0.8$ | $58.3 \pm 0.7$ | $55.0 \pm 0.3$ |
| | Linear WV | $\mathbf{39.0 \pm 0.0}$ | $61.3 \pm 0.0$ | $67.1 \pm 0.4$ | $60.0 \pm 1.2$ | $59.0 \pm 2.9$ | $\mathbf{57.3 \pm 0.6}$ |
| | Logit WV | $37.7 \pm 0.0$ | $\mathbf{61.7 \pm 0.0}$ | $67.0 \pm 0.0$ | $\mathbf{60.6 \pm 0.8}$ | $58.6 \pm 3.5$ | $57.1 \pm 0.7$ |

## B.5 RESOLVE THE DISTRIBUTION INCONSISTENCY GAP AMONG QUESTIONS

We conducted an additional experiment using pseudo-PRM scores sampled from identical distributions for correct and incorrect responses. This simulates the behavior of our aggregation methods assuming the distribution inconsistency is resolved. As shown in Table 8, when PRM scores follow a consistent distribution independent of the specific question and response, the KDE method

Table 6: Aggregating the self-rewarded verification signal from CISC.

| Aggregation Methods | Accuracy |
|---|---|
| BoN | 36.7 |
| MV | 61.3 |
| Vanilla WV | 61.3 |
| KDE WV | 61.3 |
| Linear WV | 61.7 |
| Logit WV | **62.0** |

Table 7: The MSE between the dataset-wise estimated PRM score function and the per-question-optimal function.

| PRMs | Mistral-7B | Qwen2.5-1.5B | Qwen2.5-7B | DeepSeek-7B | DeepSeek-1.5B |
|---|---|---|---|---|---|
| Qwen-PRM800K-7B | 0.165 | 0.173 | 0.361 | 0.219 | 0.108 |
| Qwen-PRM-7B | 0.132 | 0.103 | 0.127 | 0.123 | 0.101 |
| Llama3.1-Mistral-8B | 0.103 | 0.123 | 0.257 | 0.182 | 0.110 |
| Llama3.1-DS-8B | 0.116 | 0.142 | 0.166 | 0.173 | 0.183 |
| Skywork-PRM-7B | 0.141 | 0.166 | 0.294 | 0.226 | 0.112 |
| Skywork-PRM-1.5B | 0.162 | 0.171 | 0.224 | 0.261 | 0.152 |
| Shepherd-Mistral-7B | 0.465 | 0.380 | 0.263 | 0.365 | 0.423 |

improves significantly, surpassing parametric aggregation approaches. This result aligns with our expectations and further validates both our analysis of KDE and our proposed theory.

## C  DISCUSSION ON CALIBRATION COST

the calibration cost is a negligible one-time investment compared to the substantial, recurring test-time savings. Our Logit Weighted Voting method matches the baseline accuracy while strictly requiring only 21.3% of the computation, effectively saving 78.7% of the compute per query. Because our calibration relies on a small set of 100 questions, **this initial computational cost is recovered after serving just 127 user queries** ($100/0.787 \approx 127$). In any practical deployment serving thousands of requests, this break-even point is reached almost immediately, rendering the calibration overhead insignificant relative to the long-term efficiency gains.

## D  LIMITATIONS AND FUTURE WORK

**Limitations.** Our work, while demonstrating significant gains, has several limitations. First, our theoretical framework relies on conditional independence assumptions which are simplifications of the complex dependencies between generated responses. Second, our proposed calibration methods learn a single, global weighting function. As our analysis in Section 4.3 shows, the truly optimal function varies on a per-question basis, and our attempts to learn a meta-model to predict these per-question functions were unsuccessful, indicating this is a non-trivial challenge. Finally, while effective, our calibration methods require a small, one-time labeled dataset, and our evaluation has been focused on the domain of mathematical reasoning.

**Future Work.** These limitations point to several promising avenues for future research. The primary challenge is to bridge the gap between global and per-question optimal weighting. Developing methods that can adapt the weighting function at test time based on question-specific features or initial response characteristics could yield further performance gains. Another direction is to explore the generalization of our calibrated aggregation framework to other domains beyond mathematics and to other TTS paradigms, such as sequential refinement or tree-of-thoughts search. Lastly, investigating semi-supervised or unsupervised calibration techniques could reduce the reliance on labeled data, making the approach more accessible and scalable.

Table 8: Performance of aggregation methods on a pseudo PRM whose score for correct and incorrect samples are drawn from a single distribution independent of the questions, respectively.

| PRM | Method | Mistral-7B | Qwen2.5-1.5B | Qwen2.5-7B | DeepSeek-1.5B | DeepSeek-7B | Average |
|---|---|---|---|---|---|---|---|
| Pseudo PRM | Optimal | 40.7 | 67.9 | 70.8 | 63.0 | 67.0 | 61.9 |
| | BoN | 33.1 | 56.1 | 62.9 | 51.1 | 57.4 | 52.1 |
| | MV | 29.0 | 59.7 | 66.0 | 53.3 | 58.3 | 53.3 |
| | Vanilla WV | 32.5 | 60.8 | 66.8 | 55.1 | 59.5 | 54.9 |
| | KDE WV | **38.3** | **62.4** | **66.9** | **57.3** | 60.4 | **57.1** |
| | Linear WV | 33.6 | 60.3 | 66.3 | 55.5 | 61.6 | 55.5 |
| | Logit WV | 33.7 | 60.3 | 66.6 | 56.5 | **61.8** | 55.8 |

**Ethics Statement.** This research aims to improve the computational efficiency of large language models, a goal with positive ethical implications. By achieving higher performance with less computational cost, our methods can contribute to reducing the energy consumption and environmental impact associated with deploying state-of-the-art AI systems. The datasets and models used in this work are standard, publicly available benchmarks and open-source models widely used by the research community. Our work does not involve human subjects, nor does it introduce new capabilities that would increase the risk of misuse of language models. On the contrary, by developing a more nuanced understanding of how to verify and aggregate machine-generated reasoning, this research could contribute to making LLMs more reliable and less prone to generating confident but incorrect outputs. All authors have read and adhered to the ICLR Code of Ethics.

**Reproducibility Statement.** We are committed to ensuring the reproducibility of our work. All LLMs and PRMs used are publicly available models, and we provide details of these models in Section 5.1. The experiments were conducted on the MATH dataset, a standard public benchmark. Our data splitting procedure for calibration and testing is described in Section 5.1. The full theoretical derivation for Theorem 3.2 is provided in Appendix A.1. Key implementation details for our proposed calibration methods, including the KDE procedure and the grid search ranges for parametric models, are described in Section 4.

**LLM Usage.** We use LLMs to polish the writing of this paper, including identifying spelling, grammar mistakes. All suggestions from the LLM are verified by the authors before being incorporated into the paper.

