# OpenReview forum: "Optimal Aggregation of LLM and PRM Signals for Efficient Test-Time Scaling"
_ICLR.cc/2026/Conference — ICLR 2026 Poster_

### Official Review · Reviewer_z2Y8 · 2025-10-26

**Soundness:** 2
**Presentation:** 3
**Contribution:** 2
**Rating:** 4
**Confidence:** 3

**Summary:**

The paper proposes a new test-time scaling algorithm for LLMs when using PRMs (process reward models). The core of the method consists of computing a score for each answer appearing in a pool of generated answers, where the score depends on an *PRM Signal Term* and a *LLM Signal Term*. The authors propose ways of estimating these terms for a given LLM-PRM pair.

In its current form, **I lean towards rejection of the paper.** As outlined below, the theoretical foundation seems imprecise, evaluation is not very extensive (only MATH/MATH500, which means it remains unclear to what degree results such as the calibration weights would transfer across datasets), and accuracy gains are marginal without a clear way of selecting one of the proposed methods in practice for a given LLM-PRM pair. However, the paper certainly proposes some interesting ideas, and if the above points are addressed, it could warrant acceptance.

**Strengths:**

The results on PRM score distributions across different LLM-PRM pairs (e.g. Figure 1) are quite interesting and could be of independent interest to the research community.

Theorem 3.2, the derivation of picking the best response in a pool of responses by computing a response-level score depending on an LLM and PRM signal, is quite interesting (yet there remain questions about the derivation, see below).

The paper is mostly well-written and clear.

**Weaknesses:**

### Theoretical Foundations
The theory is imprecise and unclear.
For example, the derivation from Bayes' theorem in Section 3.1 does not make sense to me. First, it is unclear how this probability $P(\alpha_k | \mathcal{G},\mathcal{P},M,V)$ is even defined (is this supposed to be the probability that $\alpha_k$ is the correct answer under the given conditioning? If so, what is the randomness over?), and why maximizing it is the desirable thing to do. Next, the authors say they can decompose the probability $P(\mathcal{G},\mathcal{P}|...)$ into $P(\mathcal{G}|...)\times P(\mathcal{P}|...)$, which would only be possible if the (conditional) probabilities of $\mathcal{G}$ and $\mathcal{P}$ were independent, which clearly does not seem to be the case.

### Experiments
The experiments do not contain any confidence intervals/error bars. This seems to be quite important, as e.g. Figure 2 shows that the entire gain in compute could very well be within the margin of error.

While the authors test out many LLM-PRM pairs, they only seem to test on MATH and MATH500, the latter of which is known to be quite easy. Results on other datasets would greatly improve the quality of the experiments. Furthermore, it is often not even clear what dataset the results are reported on (e.g. Table 1 does not contain any information on this in the text or caption. From the previous paragraph, it is likely either MATH or MATH500, but it is unclear which one of these). In particular, it is completely unclear if any of the calibrated weights would transfer across datasets, which, in the current version of the paper, makes it seem like this is an entirely dataset-dependent method.

An explanation of compute tradeoffs would help evaluate the proposed method better. In particular, in order to compute the calibrated weights, a calibration set (with LLM responses) has to be generated. This seems to be quite expensive in practice.

The results show that accuracy gains can be obtained over baselines like best-of-n (BoN) or majority voting (MV), but these gains are not very consistent, neither across LLM-PRM pairs, nor across the proposed methods (KDE vs Linear vs Logit); it remains unclear how to pick the best method for each LLM-PRM pair in practice. Furthermore, the improvement gains are often not very significant, and sometimes the proposed method is even outperformed by the baselines (cf. Table 1).

To summarize, the paper proposes an interesting approach to test-time scaling that's certainly worth exploring more. However, the main drawbacks I see are a) questionable theoretical derivation (Section 3.1); b) limited experimental validation (just one dataset, no results on compute overhead of the proposed method); c) marginal accuracy gains in practice.

**Questions:**

- lines 121-124: Are $s_i$ and $\alpha_i$ the same thing? Or are they different? In particular, if they are the same, then assuming a uniform prior over answers doesn't really make sense; if, on the other hand, the $\alpha_i$ are supposed to constitute the set of *all* possible answers, then it doesn't seem to make sense to assume this set is finite.
- what do the red dots in Figure 1 represent? This should be explained in the figure caption.
- is the probability $q_M$ (line 165) assumed to be specific to a prompt (i.e. varies across prompts)?
- Some of the figure fonts are hard to read (e.g. Figure 2, 3).

---

> ### Author Response · Authors · 2025-11-19
> **Author Rebuttal for Reviewer z2Y8 - Part 1**
>
> We thank Reviewer z2Y8 for their careful and detailed reading of our paper. We are glad that the reviewer found our empirical and theoretical results on different LLM-PRM pairs interesting.
>
> We address each weakness and question below:
>
> ## Weakness: Clarify theoretical foundations
>
> We appreciate the reviewer's request for clarification here.
>
> * On $P(\alpha_{k}|\mathcal{G},\mathcal{P},M,V)$: This term is the posterior probability that $\alpha_k$ is the true, correct answer, given all the evidence we have: the set of generated responses ($\mathcal{G}$), the set of PRM scores ($\mathcal{P}$), and the models themselves (M, V). The "randomness" is over the true answer. Maximizing this posterior probability is the standard Maximum a Posteriori (MAP) estimation objective, a classical Bayesian approach.
> * On $P(\mathcal{G},\mathcal{P}|...)$ decomposition: The reviewer is absolutely correct that $\mathcal{G}$ and $\mathcal{P}$ are not independent. Our derivation explicitly models the dependency that the PRM scores $\mathcal{P}$ is dependent on the generation $\mathcal{G}$. We use the chain rule of probability, which reflects the causal process: $P(\mathcal{G},\mathcal{P}|\dots) = P(\mathcal{P} | \mathcal{G}, \alpha_{k}, M, V) \times P(\mathcal{G} | \alpha_{k}, M, V)$. Specifically, the first decomposed term is conditioned on $\mathcal{G}$, reflecting such dependency. In conclusion, this decomposition does not assume independence and is a standard application of the chain rule. We have revised Section 3.1 to make this step and the causal logic more explicit.
>
> ## Weakness: Confidence intervals
>
> This is a valid point. To address the statistical significance of the results, we run the experiments in a new setting to introduce randomness in the calibration set. Specifically, for the MATH dataset, we select the first 2k questions as the pool for calibration, and randomly sample 1k questions from the pool as the calibration set for each run. Similarly, for MATH500, the calibration pool size is 200, and we randomly sample 100 questions from the pool as the calibration set. Questions outside of the pool are used as the test set. We run each experiment 3 times with different random seeds and report the mean and standard deviation of the accuracy. We have updated the results in the revised paper accordingly. Here we provide the results averaged across the LLMs below. As we can see, **the standard deviation is much smaller than the performance gain, demonstrating its statistical significance**.
>
> | **Method**| **Qwen-PRM800K-7B** | **Qwen-PRM-7B** | **Llama3.1-Mistral-8B** | **Llama3.1-DS-8B** | **Skywork-PRM-1.5B** |
> | - | - | - | - | - | - |
> | Optimal | 61.7 ± 0.0| 66.5 ± 0.0| 65.3 ± 0.0| 65.0 ± 0.0 | 68.4 ± 0.0 |
> | BoN | 52.7 ± 0.0| 61.8 ± 0.0| 58.3 ± 0.0| 58.9 ± 0.0 | 65.3 ± 0.0 |
> | MV| 57.1 ± 0.0| 57.1 ± 0.0| 57.1 ± 0.0| 57.1 ± 0.0 | 59.2 ± 0.0 |
> | Vanilla WV| 57.4 ± 0.0| 60.8 ± 0.0| 59.4 ± 0.0| 59.7 ± 0.0 | 63.5 ± 0.0 |
> | **KDE WV**| 57.5 ± 0.0| 60.2 ± 0.2| 60.2 ± 0.2| 59.7 ± 0.2 | 63.2 ± 0.5 |
> | **Linear WV** | 57.6 ± 0.0| 62.9 ± 0.0| 61.9 ± 0.2| 61.1 ± 0.1 | **66.0 ± 0.1** |
> | **Logit WV**| **57.6 ± 0.0**| **63.3 ± 0.0**| **62.6 ± 0.1**| **61.5 ± 0.0** | 65.9 ± 0.0 |
>
> ## Weakness: More datasets
>
> We agree that experiments on more diverse datasets can greatly improve the quality of the experiments. To demonstrate broader applicability, we further experimented on a wide range of tasks beyond math. In the MMLU-Pro-CoT-Eval dataset, there are approximately 150 questions in each task, among which 50 are used as the calibration set and the rest as the test set. Llama-3.1-8B-Instruct and VersaPRM are the LLM-PRM pair. As seen in the following table, **the proposed methods are consistently effective on a wide range of tasks beyond the math domain**. We believe those experiments on a wide variety of domains indeed make the experiments much more comprehensive, thanks to the reviewer’s advice.
>
> | **Method**| **Health** | **CS** | **Econ** | **Chem** | **Other** | **Physics** | **Law**| **Eng**| **History** | **Math** | **Phil** | **Average** |
> | - | - | - | - | - | - | - | - | - | - | - | - | - |
> | Optimal | 71.8 | 78.0 | 76.0 | 85.0 | 79.0| 84.0| 65.3 | 78.0 | 62.0| 85.0 | 70.7 | 75.9|
> | BoN | 61.2 | 57.0 | 62.0 | 64.0 | 61.0| 62.0| 45.3 | 55.0 | 46.0| 63.0 | 50.5 | 57.0|
> | MV| 62.4 | 55.0 | 61.0 | 64.0 | 57.0| 62.0| 35.8 | 56.0 | 44.0| 61.0 | 42.4 | 54.6|
> | Vanilla WV| 63.5 | 57.0 | 62.0 | 66.0 | 59.0| 64.0| 43.2 | 60.0 | 46.0| 65.0 | 48.5 | 57.7|
> | **KDE WV**| **64.7** | 57.0 | 63.0 | **66.0** | 58.0| 64.0| 42.1 | 58.0 | 45.0| 65.0 | 46.5 | 57.2|
> | **Linear WV** | 63.5 | 57.0 | **64.0** | 66.0 | **62.0**| **67.0**| 44.2 | **63.0** | **47.0**| **66.0** | 46.5 | 58.7|
> | **Logit WV**| 62.4 | **58.0** | 64.0 | 65.0 | 60.0| 66.0| **45.3** | 63.0 | 44.0| 66.0 | **53.5** | **58.8**|

---

> ### Author Response · Authors · 2025-11-19
> **Author Rebuttal for Reviewer z2Y8 - Part 2**
>
> ## Weakness: Result description
>
> We apologize for the omission. Table 1 reports results on the MATH dataset (the 6.5k sample test split). Figure 2 (Left) shows MATH, and Figure 2 (Right) shows MATH500. We have added this information clearly to all captions and text in the revision.
>
> ## Weakness: Cross-dataset transfer
>
> To further examine the effectiveness of calibration sets that are not in the same domain as the test set, we calibrate an aggregator on each of the tasks, and report its **average accuracy when tested on the other tasks** in the following table, where each column correspond to the calibrator trained on that task. As we can see, **despite being calibrated on various “Out-of-domain” calibration sets, the proposed methods consistently outperform the baselines, showing strong transferability**.
>
> | **Method**| **Chem** | **CS** | **Econ** | **Eng**| **Health** | **History** | **Law**| **Math** | **Other** | **Phil** | **Physics** | **Average** |
> | - | - | - | - | - | - | - | - | - | - | - | - | - |
> | Optimal | 74.8 | 75.2 | 75.5 | 75.3 | 75.7 | 76.5| 76.8 | 74.8 | 75.3| 76.0 | 74.8| 75.5|
> | BoN | 56.4 | 56.8 | 56.5 | 57.2 | 56.8 | 57.7| 58.2 | 56.5 | 56.6| 58.0 | 56.4| 57.0|
> | MV| 53.7 | 54.3 | 53.9 | 54.3 | 53.3 | 55.1| 56.3 | 53.8 | 54.2| 55.5 | 53.5| 54.4|
> | Vanilla WV| 57.0 | 57.5 | 57.3 | 57.5 | 56.8 | 58.4| 59.1 | 57.0 | 57.5| 58.7 | 56.8| 57.6|
> | **KDE WV**| 57.5 | 56.3 | 55.0 | 57.7 | 55.0 | 57.5| 59.8 | 56.9 | 56.3| 57.8 | 57.4| 57.0|
> | **Linear WV** | **57.7** | **57.9** | **58.1** | 57.7 | 55.2 | **59.5**| 58.9 | 57.1 | **58.0**| 58.4 | 56.7| 57.8|
> | **Logit WV**| 57.7 | 57.8 | 57.9 | **58.1** | **57.8** | 59.4| **59.9** | **57.5** | 57.9| **58.8** | **57.7**| **58.2**|
>
> ## Weakness: Calibration cost
>
> We appreciate the reviewer’s feedback and have clarified that the calibration cost is a negligible one-time investment compared to the substantial, recurring test-time savings. Our Logit Weighted Voting method matches the baseline accuracy while strictly requiring only 21.3% of the computation, effectively saving 78.7% of the compute per query. Because our calibration relies on a small set of 100 questions, **this initial computational cost is recovered after serving just 127 user queries** ($100 / 0.787 \approx 127$). In any practical deployment serving thousands of requests, this break-even point is reached almost immediately, rendering the calibration overhead insignificant relative to the long-term efficiency gains.
>
> ## Weakness: Inconsistent/marginal gains
>
> We appreciate the reviewer raising this point, as it allows us to better contextualize the magnitude of our results. While we understand that the numerical gains might appear modest at first glance, we believe they represent a meaningful advancement for a few key reasons:
>
> - **Consistency across models:** Our methods (Logit WV, Linear WV) consistently improve upon the baselines (MV, BoN, Vanilla WV) across the vast majority of the 35 model pairs we tested. For instance, **Logit WV achieves an average accuracy of 62.18 compared to the best baseline (Vanilla WV) at 60.16—a gain of 2.02 points with a very low standard deviation of 0.02**. We believe this consistency suggests the method is reliable rather than noisy.
> - **Efficiency in the context of scaling:** In the specific context of Test-Time Scaling, gains often suffer from diminishing returns, where **achieving a linear increase in accuracy typically requires an exponential increase in test-time sampling cost** [1, 2]. Because our method improves the maximum accuracy without requiring additional samples, we view this 2% gain as a significant efficiency boost that would otherwise be very expensive to achieve via raw compute.
> - **Robustness and Cost:** We also observed that this advantage holds even under challenging domain shifts (**achieving a 0.6 gain even when calibration and test domains differ significantly**). Given that our calibration method introduces **negligible computational overhead**, as discussed earlier, we believe this "free" improvement offers a highly favorable cost-benefit ratio for practical applications.
>
> [1] Scaling LLM Test-Time Compute Optimally can be More Effective than Scaling Model Parameters, Snell et al., ICLR 2025
>
> [2] Inference Scaling for Long-Context Retrieval Augmented Generation, Yue et al., ICLR 2025

---

> ### Author Response · Authors · 2025-11-19
> **Author Rebuttal for Reviewer z2Y8 - Part 3**
>
> ## Question 1: Explain $s_i$ vs $\alpha_k$
>
> We thank the reviewer for pointing out the unclarity. $s_i$ and $\alpha_k$ are different.
>
> * $s_i$ is the final answer (a string) from a single generation $g_i$.
> * $\mathcal{A} = \{\alpha_1, ..., \alpha_m\}$ is the set of unique answers present in the pool of L generations. For example, if we generate L=10 responses, we might get 7 "A"s and 3 "B"s. Here, $s_1$="A", ..., $s_7$="A", $s_8$="B", etc. The set of unique answers is $\mathcal{A}$ = {$\alpha_1, \alpha_2$} = {"A", "B"}. This set is finite for any given problem. We have clarified this notation in our revision.
>
> ## Question 2: Explain red dots in Figure 1
>
> We apologize for the missing explanation. The red dot in Figure 1 is the zero-crossing point of the weighting functions, and the horizontal dashed line indicates the zero weight. We mark the zero crossing point to highlight the presence of negative weights. We have clarified them in the figure caption in our revision.
>
> ## Question 3: Explain $q_M$ implementation
>
> Ideally, $q_M$ (the LLM's reliability) is prompt-specific.
>
> Our 'Optimal' method (Section 3.2), which requires ground-truth labels, does estimate this on a per-question basis.
>
> However, for our practical calibration methods (Section 4), we have no access to the true $q_M$ for a new test question, as it would require the ground truth answer. Our KDE WV method approximates it using the calibrated probabilities of the N responses, and our parametric methods (Logit WV) learn a global function that works well on average across all $q_M$ values. We have clarified this distinction in our revised paper.
>
> ## Question 4: Adjust figure fonts
>
> Thank you. We have increased the font size in Figure 2 and 3 for readability.
>
>
>
> Again, we’d like to thank the reviewer for their detailed review and valuable advice, **all of which is incorporated in the revised paper marked in blue for your review**.

---

> ### Author Response · Authors · 2025-11-27
> **Follow-up on our response**
>
> Dear Reviewer z2Y8,
>
> We thank you again for the time and effort you dedicated to reviewing our paper.
>
> We wanted to gently follow up to ensure you have had a chance to read our rebuttal. We believe we have addressed your main concerns, particularly regarding theoretical foundations, additional datasets, and transferability.
>
> As the discussion period is coming to a close soon, we are eager to hear your thoughts. If you have any remaining questions or require further clarification, please let us know so we can respond before the deadline.
>
> Best regards, The Authors

---

### Official Review · Reviewer_rGBo · 2025-10-31

**Soundness:** 3
**Presentation:** 4
**Contribution:** 2
**Rating:** 4
**Confidence:** 4

**Summary:**

This paper investigates into the utilization of the Process Reward Models (PRMs) in test time scalings (TTS) for LLMs. The authors argue that simple majority voting sometimes outperforms the PRM guided method (such as Best-of-N selection) which indicates the usage of PRM is sub-optimal in TTS. The authors propose a framework with Maximum a Posteriori (MAP) estimation and demonstrate the optimal aggregation method for TTS is via weighted majority vote (with PRM). Besides the theoretical framework, they also propose two practical methods based on KDE and parametric learning to learn these weighting functions from a small labeled dataset. They empirically show the proposed method can achieve similar accuracy as baselines while using only 21.3%-37.1% of the computes, on 5 LLMs and 7 PRMs.

**Strengths:**

1. In general the paper is well written and the presentation is pretty clear. The proposed theoretical framework (i.e., the MAP based method)  is formulated in a clean way and mathematically makes sense.

2. The experiments are quite comprehensive. The authors verified their claims across 35 LLM-PRM pairs and demonstrate consistent results on these models and datasets.

3. The experiments show promising results. The proposed method is able to achieve similar accuracy as other baselines with only 21.3%-37.1% of the compute, which will be somewhat useful in the real world applications.

**Weaknesses:**

Despite the clear presentation and the clean mathematical formulation, I have the following concerns about the paper:

1. The independence assumption (i.e., assumption 3.1) is quite strong. Assumption 3.1 assumes that PRM scores and the LLM generations are conditionally independent of each other given the true answer, the verifier and the model. I would argue that both LLM generation and the PRM scores are very unlikely independent of each other, instead, the LLM is likely to generate responses that are "correlated" with each other. This assumption is critical, however, the paper doesn't discuss/validate it empirically, neither do it discuss the potential impact on the conclusion.

2. The ablation study is not sufficient, especially on the calibration dataset. There are a few critical questions that can impact the generalizability/applicability of the method: 1/ How does the distribution shift between calibration and test set affect the performance? How much can the calibration results be transferred to a different domain? 2/ Even within the same domain (i.e., math), how much does the calibration results transfer between difficulty of the problem, or formulation of the problem? So on and so forth.

3. Missing statistical significance: The authors report absolute results in table 1 and the best numbers are flagged in bold. However, the score gaps with other baseline are usually very small, I would strongly recommend to include standard deviation or confidence intervals to indicate whether these improvements are statistically significant.

**Questions:**

1. How sensitive is the method to the violation of the Assumption 3.1? I am wondering whether you have some empirical data points to show the correlation between responses or the PRM scores and see how does this correlation impact the final performance?

2. Did you perform ablation study to see how the calibration set domain change can impact the test set accuracy? I would like to understand the domain transferability of the method. It can be truly different domain, different types of math problems or even the difficulty of the same type of math problems.

3. I think we are missing some baselines, have we compared with other type of methods, such as in [1] and [2]? To be fair, the baselines we compared in the paper are quite simple. [2] is a bit complicated as they require to train a model with RL, however, the method in [1]
 is very straightforward and not necessarily need the calibration dataset.

[1] Confidence Improves Self-Consistency in LLMs, Taubenfeld et al., ACL 2025
[2] The Majority is not always right: RL training for solution aggregation, Zhao et al.

---

> ### Author Response · Authors · 2025-11-19
> **Author Rebuttal for Reviewer rGBo - Part 1**
>
> We thank Reviewer rGBo for their thorough review, complimenting our clear presentation, clean theoretical formulation, and comprehensive experiments. We also appreciate the critical questions, which will help us strengthen the paper significantly.
>
> We address each weakness and question below:
>
> ## Weakness 1 & Question 1: The independence assumption
>
> We appreciate the reviewer’s scrutiny of Assumption 3.1. We agree that without specific conditioning, LLM generations and PRM scores would indeed exhibit statistical dependence. However, we respectfully argue that within our theoretical framework, the independence assumption is statistically valid **precisely because we explicitly condition on the model along with its generation sampling distribution characterized by $q_M$**.
>
> For LLM generations, in Section 3.1, we model the generation process as sampling from a parameterized distribution where the probability of the correct answer is $q_M$ and incorrect answers are distributed over the remaining space. By conditioning on these fixed parameters, the generations $g_i$ become **Independent and Identically Distributed (i.i.d.)** draws from this specific distribution. In this case, the challenge of applying the theory in practice is not the violation of the assumptions, but rather reliably estimating $q_M$, which we have discussed in Section 5.3.
>
> Similarly, for the PRM scores, we assume the scores are drawn from specific **class-conditional distributions**: $P(p|c=1, V)$ for correct responses and $P(p|c=0, V)$ for incorrect ones. Once the correctness $c_i$ is conditioned upon, the PRM score $p_i$ is treated as an independent sample from the corresponding distribution.
>
> To sum up, strictly within the context of our Maximum Likelihood Estimation (Eq. 1), the independence assumption holds because we are maximizing over a fixed set of parameters that fully define the underlying probability distributions. This formulation allows us to derive the optimal weighting function.
>
> ## Weakness 2 & Question 2: Ablation studies
>
> We agree that more ablation studies, especially those on the calibration set, can largely strengthen our paper. We have added the following new ablations in our revised paper.
>
> 1. **Effectiveness in domains beyond math.** To demonstrate broader applicability, we further experimented on a wide range of tasks beyond math. In the MMLU-Pro-CoT-Eval dataset, there are approximately 150 questions in each task, among which 50 are used as the calibration set and the rest as the test set. Llama-3.1-8B-Instruct and VersaPRM are the LLM-PRM pair. As seen in the following table, **the proposed methods are consistently effective on a wide range of tasks beyond the math domain**.
>
> |**Method**|**Health**|**CS**|**Econ**|**Chem**|**Other**|**Physics**|**Law**|**Eng**|**History**|**Math**|**Phil**|**Average**|
> |-|-|-|-|-|-|-|-|-|-|-|-|-|
> |Optimal|71.8|78.0|76.0|85.0|79.0|84.0|65.3|78.0|62.0|85.0|70.7|75.9|
> |BoN|61.2|57.0|62.0|64.0|61.0|62.0|45.3|55.0|46.0|63.0|50.5|57.0|
> |MV|62.4|55.0|61.0|64.0|57.0|62.0|35.8|56.0|44.0|61.0|42.4|54.6|
> |Vanilla WV|63.5|57.0|62.0|66.0|59.0|64.0|43.2|60.0|46.0|65.0|48.5|57.7|
> |**KDE WV**|**64.7**|57.0|63.0|66.0|58.0|64.0|42.1|58.0|45.0|65.0|46.5|57.2|
> |**Linear WV**|63.5|57.0|64.0|**66.0**|**62.0**|**67.0**|44.2|63.0|**47.0**|66.0|46.5|58.7|
> |**Logit WV**|62.4|**58.0**|**64.0**|65.0|60.0|66.0|**45.3**|**63.0**|44.0|**66.0**|**53.5**|**58.8**|
>
> 2. **Effectiveness of Out-of-domain calibration sets / Transferability of the calibration results.** To further examine the effectiveness of calibration sets that are in *totally different domain* than the test set, we calibrate an aggregator on each of the tasks, and report its **average accuracy when tested on the other tasks** in the following table, where each column correspond to the calibrator trained on that task. As we can see, **despite being calibrated on various “Out-of-domain” calibration sets, the proposed methods consistently outperform the baselines, showing strong transferability**.
>
> |**Method**|**Chem**|**CS**|**Econ**|**Eng**|**Health**|**History**|**Law**|**Math**|**Other**|**Phil**|**Physics**|**Average**|
> |-|-|-|-|-|-|-|-|-|-|-|-|-|
> |Optimal|74.8|75.2|75.5|75.3|75.7|76.5|76.8|74.8|75.3|76.0|74.8|75.5|
> |BoN|56.4|56.8|56.5|57.2|56.8|57.7|58.2|56.5|56.6|58.0|56.4|57.0|
> |MV|53.7|54.3|53.9|54.3|53.3|55.1|56.3|53.8|54.2|55.5|53.5|54.4|
> |Vanilla WV|57.0|57.5|57.3|57.5|56.8|58.4|59.1|57.0|57.5|58.7|56.8|57.6|
> |**KDE WV**|57.5|56.3|55.0|57.7|55.0|57.5|59.8|56.9|56.3|57.8|57.4|57.0|
> |**Linear WV**|**57.7**|**57.9**|**58.1**|57.7|55.2|**59.5**|58.9|57.1|**58.0**|58.4|56.7|57.8|
> |**Logit WV**|57.7|57.8|57.9|**58.1**|**57.8**|59.4|**59.9**|**57.5**|57.9|**58.8**|**57.7**|**58.2**|

---

> ### Author Response · Authors · 2025-11-19
> **Author Rebuttal for Reviewer rGBo - Part 2**
>
> ## Weakness 3: Missing statistical significance
>
> This is a valid point. To address the statistical significance of the results, we run the experiments in a new setting to introduce randomness in the calibration set. Specifically, for the MATH dataset, we select the first 2k questions as the pool for calibration, and randomly sample 1k questions from the pool as the calibration set for each run. Similarly, for MATH500, the calibration pool size is 200, and we randomly sample 100 questions from the pool as the calibration set. Questions outside of the pool are used as the test set. We run each experiment 3 times with different random seeds and report the mean and standard deviation of the accuracy. We have updated the result in the revised paper accordingly. We provide the results averaged across the LLMs below. As we can see, **the standard deviation is much smaller than the performance gain, demonstrating its statistical significance**.
>
> |**Method**|**Qwen-PRM800K-7B**|**Qwen-PRM-7B**|**Llama3.1-Mistral-8B**|**Llama3.1-DS-8B**|**Skywork-PRM-1.5B**|
> |-|-|-|-|-|-|
> |Optimal|61.7 ± 0.0|66.5 ± 0.0|65.3 ± 0.0|65.0 ± 0.0|68.4 ± 0.0|
> |BoN|52.7 ± 0.0|61.8 ± 0.0|58.3 ± 0.0|58.9 ± 0.0|65.3 ± 0.0|
> |MV|57.1 ± 0.0|57.1 ± 0.0|57.1 ± 0.0|57.1 ± 0.0|59.2 ± 0.0|
> |Vanilla WV|57.4 ± 0.0|60.8 ± 0.0|59.4 ± 0.0|59.7 ± 0.0|63.5 ± 0.0|
> |**KDE WV**|57.5 ± 0.0|60.2 ± 0.2|60.2 ± 0.2|59.7 ± 0.2|63.2 ± 0.5|
> |**Linear WV**|57.6 ± 0.0|62.9 ± 0.0|61.9 ± 0.2|61.1 ± 0.1|**66.0 ± 0.1**|
> |**Logit WV**|**57.6 ± 0.0**|**63.3 ± 0.0**|**62.6 ± 0.1**|**61.5 ± 0.0**|65.9 ± 0.0|
>
> ## Question 3: Additional baselines
>
> Thank you for these references. CISC is an approach that uses the confidence of the LLM as verification signals. While the verification signal is not from a PRM, our aggregation method is still applicable to CISC to utilize the signal more effectively. Here, we use the best-performing “P(True)” implementation of CISC to examine the effectiveness of the proposed method in aggregating self-rewarded verification signals. As we can see from the results on the MATH500 dataset below, **the proposed method can more effectively aggregate the self-rewarded verification signal than vanilla weighted majority voting and Best-of-N**. We have incorporated the experiments and discussions on CISC in our revised paper.
>
> |**Aggregation Methods**|**Accuracy**|
> |-|-|
> |BoN|36.7|
> |MV|61.3|
> |Vanilla WV|61.3|
> |**KDE WV**|61.3|
> |**Linear WV**|61.7|
> |**Logit WV**|**62.0**|
>
>
>
> Again, we’d like to thank the reviewer for their detailed review and valuable advice, **all of which is incorporated in the revised paper marked in blue for your review**.

---

> ### Author Response · Authors · 2025-11-27
> **Follow-up on our response**
>
> Dear Reviewer rGBo,
>
> We thank you again for the time and effort you dedicated to reviewing our paper.
>
> We wanted to gently follow up to ensure you have had a chance to read our rebuttal. We believe we have addressed your main concerns, particularly regarding independent assumptions and ablation studies.
>
> As the discussion period is coming to a close soon, we are eager to hear your thoughts. If you have any remaining questions or require further clarification, please let us know so we can respond before the deadline.
>
> Best regards, The Authors

---

### Official Review · Reviewer_hEsP · 2025-10-31

**Soundness:** 3
**Presentation:** 2
**Contribution:** 3
**Rating:** 6
**Confidence:** 2

**Summary:**

This paper focuses on how to leverage the signals from PRMs (process reward models) in test-time scaling. They show empirical results with an optimal weighting function, incl. practical calibration methods to learn these functions. They show that using these methods improves test-time-scaling efficiency.

**Strengths:**

The paper presents results across 5 LLMs and 7 PRMs and they show that this helps boost test-time scaling efficiency. They also present different aggregations (see Table 1) which shows better results across multiple model-PRM pairs and the MATH datasets.  They achieve that with way less computation (see line 71 and 306 for details).

They also compare with multiple baselines, such as majority voting, best of N, vanilla weighted vote. Their weighting function calibration significantly improves TTS efficiency.

They also showcase how negative weights are necessary in using PRM signals

**Weaknesses:**

The paper presents very simple concepts in the abstract but does it in a way by introducing nomenclature that makes it hard to understand. For example, it is not clear why PRMs are important for TTS and their applicability. I would start with that, then present the actual efforts.

**Questions:**

The MATH dataset is from 2021, do we think that this approach will generalize to novel unseen benchmarks?

---

> ### Author Response · Authors · 2025-11-19
> **Author Rebuttal for Reviewer hEsP**
>
> We thank Reviewer hEsP for their positive feedback on our comprehensive experiments and the effectiveness of our proposed methods. We appreciate the reviewer's suggestions for improving the paper's clarity and generalizability.
>
> We address each point below:
>
> ## Weakness: Clarity of the abstract:
>
> We thank the reviewer for this feedback, and we agree that the paper's motivation can be stated more clearly. In our revision, we revised the abstract. We first explicitly state the foundational role of PRMs in Test-Time Scaling. As suggested by the reviewer, only after establishing their importance did we introduce the core problem: recent benchmarks show that suboptimal use of these expensive PRMs can be outperformed by simple majority voting, motivating our work to find a principled and intelligent aggregation strategy.
>
> ## Question: Generalization beyond the MATH dataset:
>
> This is an important question regarding the generalizability of our findings. To demonstrate broader applicability, we further experimented on a wide range of tasks beyond math. In the MMLU-Pro-CoT-Eval dataset, there are approximately 150 questions in each task, among which 50 are used as the calibration set and the rest as the test set. Llama-3.1-8B-Instruct and VersaPRM are the LLM-PRM pair. As seen in the following table, **the proposed methods are consistently effective on a wide range of tasks beyond the math domain**.
>
> | **Method** | **Health** | **CS**| **Econ** | **Chem** | **Other** | **Physics** | **Law**| **Eng**| **History** | **Math** | **Phil** | **Average** |
> | - | - | - | - | - | - | - | - | - | - | - | - | - |
> | Optimal | 71.8 | 78.0| 76.0| 85.0| 79.0| 84.0| 65.3| 78.0| 62.0| 85.0| 70.7| 75.9|
> | BoN| 61.2 | 57.0| 62.0| 64.0| 61.0| 62.0| 45.3| 55.0| 46.0| 63.0| 50.5| 57.0|
> | MV| 62.4 | 55.0| 61.0| 64.0| 57.0| 62.0| 35.8| 56.0| 44.0| 61.0| 42.4| 54.6|
> | Vanilla WV | 63.5 | 57.0| 62.0| 66.0| 59.0| 64.0| 43.2| 60.0| 46.0| 65.0| 48.5| 57.7|
> | **KDE WV** | **64.7**| 57.0| 63.0| **66.0** | 58.0| 64.0| 42.1| 58.0| 45.0| 65.0| 46.5| 57.2|
> | **Linear WV** | 63.5 | 57.0| **64.0** | 66.0| **62.0**| **67.0** | 44.2| **63.0** | **47.0** | **66.0** | 46.5| 58.7|
> | **Logit WV**| 62.4 | **58.0** | 64.0| 65.0| 60.0| 66.0| **45.3** | 63.0| 44.0| 66.0| **53.5** | **58.8** |
>
>
>
> Again, we’d like to thank the reviewer for their detailed review and valuable advice, **all of which is incorporated in the revised paper marked in blue for your review**.

---

> > ### Comment · Reviewer_hEsP · 2025-11-20
> >
> > The author response addresses my concern about generalization beyond the tasks and datasets used in the paper. The method was further evaluated in the MMLU-Pro-CoT-Eval dataset.

---

> > > ### Author Response · Authors · 2025-11-23
> > >
> > > Thank you for your response! We are glad to hear that the concern has been addressed.
> > >
> > > We are more than willing to address any further questions or concerns from the reviewer. If there are no further concerns, we appreciate the reviewer considering raising our score in the evaluation.

---

> > > > ### Comment · Reviewer_hEsP · 2025-11-27
> > > >
> > > > I have raised my score

---

### Official Review · Reviewer_suxy · 2025-10-31

**Soundness:** 3
**Presentation:** 3
**Contribution:** 3
**Rating:** 6
**Confidence:** 3

**Summary:**

In this work, the authors propose a MAP framework for using Process Reward Model signal to guide test time scaling in LLMs on mathematical reasoning tasks. Through theoretical analysis, they propose a weighting over completions based on a PRM signal and LLM signal, which better incorporates PRM signal into test time scaling than baselines. Importantly, they find that allowing for negative weights on completions with low PRM scores and also calibrating to the expected correctness of the LLM help improve the efficiency of test time scaling.

**Strengths:**

* The authors have an intuitive and clear theoretical framework guiding their proposed method.
* Through experiments, they demonstrate that considering this theoretical framework can help practitioners make test time scaling more efficient.

**Weaknesses:**

* After introducing the MAP model and proposing a way to estimate it using KDE, the empirical results find that logit weighing with grid search performs as well if not better than KDE. This gap makes me wonder why KDE is necessary in the first place, and if it is less necessary, calls into question the utility of the model. The authors do note that this drop in performance could be attributed to error in estimating $q_M$, but they do not validate this in the paper. I would be interested in seeing an ablation using ground truth $q_M$'s to validate the KDE approach and the theoretical model. On this note, I would also like to see elaboration on the sentence "In conclusion, accurately estimating either the PRM or the LLM part of the weight requires nuanced estimation for individual questions, explaining why the non-parametric KDE estimation underperforms the parametric ones."


Notation:
 * When you say "surpasses the performance of ... with approximately 37.1% and 21.3% of compute of compute", I think this needs to be tweaked gramatically. Also, I would say "Test time compute" instead of "compute" as to not confuse readers about other axes of compute such as FLOPs or training time. My immediate thought was that compute should be the same because you run the same number of forward passes through the PRM, for a fixed N but now I understand that the point is you can get the same accuracy with fewer N.

**Questions:**

* What is the objective used to calibrate the parametric objectives? Is it correctness on the calibration set?
* In Figure 4, I would like to see the MSE between the dataset estimated PRM score and the per-question-optimal for the whole dataset, this might be more convincing than having 5 picked questions as examples.

---

> ### Author Response · Authors · 2025-11-19
> **Author Rebuttal for Reviewer suxy**
>
> We sincerely thank Reviewer suxy for the constructive feedback. We are glad to hear that you found our theoretical framework "intuitive and clear" and appreciated the efficiency improvements demonstrated in our experiments.
>
> We have addressed your specific questions and comments below:
>
> ## Weaknesses: Explain the gap between KDE and Logit Weighting
>
> We agree that the performance gap between the parametric Logit WV and the non-parametric KDE WV is a crucial finding that warrants further discussion. While our theoretical model (Eq. 9) defines the optimal strategy1, its practical success depends on accurately estimating two terms: the PRM signal (log-likelihood ratio) and the LLM signal ($q_M$). We elaborate on the reviewer’s questions about KDE as follows:
>
> * **Why KDE was included:** We included KDE because it directly implements the optimal weighting function by estimating the required terms globally. In theory, KDE allows for more complex, non-linear weighting functions compared to fixed parametric shapes.
> * **Why KDE underperforms**: In practice, however, the flexibility of KDE often leads to noisy weighting functions due to difficulties in accurately estimating the LLM ($q_M$) and PRM terms for individual questions. For the PRM part, it decides the general shape of the weighting function. While KDE is capable of approximating complex shapes rather than fixed parametric shapes, such a capability could lead to noisy and less smooth functions in practice, which may result in degraded generalization ability. For the LLM part $q_M$, the estimation is noisy as the true value varies largely between questions.
> * **Validation of the theoretical model**: To confirm that the performance gap stems from estimation noise rather than a flaw in the theory, we performed an ablation study suggested by your feedback. We add two baselines, KDE (Opt LLM term) and KDE (Opt PRM term), which substitute the LLM and PRM terms of KDE WV with optimal estimations, respectively. As shown in **Table 2 in the revised paper**, using optimal estimation for the LLM and PRM term both lead to significantly better results, **supporting the paper's claim that the gap between KDE and "Optimal" is due to the bottleneck of practical estimation, rather than a flaw in the theory**. We provide an averaged result of Table 2 across PRMs for your review.
>
> |Method|Mistral-7B|Qwen2.5-1.5B|Qwen2.5-7B|DeepSeek-1.5B|DeepSeek-7B|Average|
> |:-|:-|:-|:-|:-|:-|:-|
> |Optimal|56.02|67.04|73.10|63.66|62.64|64.48|
> |KDE WV (Opt PRM term)|53.50|64.38|70.18|58.44|58.96|61.10|
> |KDE WV (Opt LLM term)|53.94|63.60|69.36|60.62|61.46|61.80|
> |KDE WV|50.88|62.16|68.04|54.66|55.62|58.30|
>
> ## Notation: Grammar tweak
>
> Thank you for pointing out the ambiguity. We have revised this sentence throughout the paper for clarity. A clearer phrasing would be: "Our method achieves the same performance as the best-performing baseline while using only 21.3% of the test-time compute."
>
> ## Question 1: Clarify Calibration objective
>
> We thank the reviewer for pointing out the unclarity. The objective used to calibrate the parametric methods (Logit WV and Linear WV) is the **accuracy on the held-out calibration set**. We perform a grid search for the offset parameter b and select the value that maximizes this accuracy. We have made this explicit in Section 4.2 of the revised paper.
>
> ## Question 2: MSE between local and global scoring function
>
> We appreciate this suggestion, as quantifying the variance adds rigor to our qualitative examples. In addition to the five qualitative examples, we have added a quantitative metric. We first scale the output range of the scoring functions to (-0.5, 0.5), and then compute the Mean Squared Error (MSE) between the dataset-wise estimated PRM score function and the per-question-optimal function, aggregated over all questions in the test set. This provides a more comprehensive view of the variance. As we can see in the table below, the MSE between the dataset-wise estimated PRM score function and the per-question-optimal function is inherently significant, which **supports our hypothesis that the optimal weighting function for each individual question varies largely, acting as one major challenge towards the optimal weighting function**.
>
> |**PRMs**|**Mistral-7B**|**Qwen2.5-1.5B**|**Qwen2.5-7B**|**DeepSeek-7B**|**DeepSeek-1.5B**|
> |-|-|-|-|-|-|
> |**Qwen-PRM800K-7B**|0.165|0.173|0.361|0.219|0.108|
> |**Qwen-PRM-7B**|0.132|0.103|0.127|0.123|0.101|
> |**Llama3.1-Mistral-8B**|0.103|0.123|0.257|0.182|0.110|
> |**Llama3.1-DS-8B**|0.116|0.142|0.166|0.173|0.183|
> |**Skywork-PRM-1.5B**|0.162|0.171|0.224|0.261|0.152|
>
>
>
> Thank you again for your valuable feedback, which we believe significantly improves the clarity and rigor of our paper and is **incorporated in the revised paper marked in blue**.

---

> ### Comment · Reviewer_suxy · 2025-11-21
> **Reviewer Response**
>
> Thank you for this response!
>
> What I interpret these results as telling me is that PRM log-likelihood is different for different responses, and LLM correctness is also different for different responses. Thus, it seems like the theoretically-optimal approach, fitting these densities, works best when you can estimate these quantities per-response rather than estimating over a calibration set (the WV approaches). Then, the empirical results show that if you can't fit optimally to each response, you might as well do Logit WV instead of some fancy KDE approach. I see this discussion on lines 407-419. I guess its beyond the scope of this work but it would be interesting to see if we can transform PRM scores to all come from the same distribution, sample-independently. I think this work, especially the contribution of the MAP and negative weights formulation, is of interest to the ICLR community and should be accepted.
>
> Quick notation question:
> * line 144, should this be $P(c_i |a_k, M)$ not $P(s_i | a_k, M)$? As you are interested in the probability of correctness. Also, if $s_i = a_k$, then this term is a little meaningless.

---

> > ### Author Response · Authors · 2025-11-23
> >
> > Thank you for your response and appreciation of our work! We address additional questions below.
> >
> > ## Additional question 1: Align PRM scores to the same distribution
> >
> > We confirm that the reviewer’s interpretation is accurate. While it is theoretically optimal to estimate the PRM likelihood and LLM correctness quantities for each question, this is infeasible in practice. Consequently, Logit WV offers a superior approach for estimating the weighting function compared to the fancy KDE approach.
> >
> > Regarding the suggestion to align PRM scores to the same distribution, we conducted an additional experiment using pseudo-PRM scores sampled from identical distributions for correct and incorrect responses. This simulates the behavior of our aggregation methods assuming the distribution inconsistency is resolved. As shown below, **when PRM scores follow a consistent distribution independent of the specific question and response, the KDE method improves significantly, surpassing parametric aggregation approaches. This result aligns with our expectations and further validates both our analysis of KDE and our proposed theory**.
> >
> > | **Method**    | **Mistral-7B** | **Qwen2.5-1.5B** | **Qwen2.5-7B** | **DeepSeek-1.5B** | **DeepSeek-7B** | **Average** |
> > | ------------- | -------------- | ---------------- | -------------- | ----------------- | --------------- | ----------- |
> > | Optimal       | 40.7           | 67.9             | 70.8           | 63.0              | 67.0            | 61.9        |
> > | BoN           | 33.1           | 56.1             | 62.9           | 51.1              | 57.4            | 52.1        |
> > | MV            | 29.0           | 59.7             | 66.0           | 53.3              | 58.3            | 53.3        |
> > | Vanilla WV    | 32.5           | 60.8             | 66.8           | 55.1              | 59.5            | 54.9        |
> > | **KDE WV**    | **38.3**       | **62.4**         | **66.9**       | **57.3**          | 60.4            | **57.1**    |
> > | **Linear WV** | 33.6           | 60.3             | 66.3           | 55.5              | 61.6            | 55.5        |
> > | **Logit WV**  | 33.7           | 60.3             | 66.6           | 56.5              | **61.8**        | 55.8        |
> >
> > Furthermore, while we agree aligning PRM scores to the same distribution is a promising direction for bridging the gap toward optimal weighting functions, achieving this via static transformations is difficult in practice, as such transformations are question- and response-agnostic and cannot account for variability in scoring distributions among questions. Implementing question-aware transformations would likely require learning an adaptive model; indeed, our prior attempts to learn a pre-question weighting function struggled to generalize, highlighting the significant challenge inherent in this approach. Alternatively, a distribution consistency objective could be directly incorporated during PRM training—a direction that merits further investigation.
> >
> > We appreciate the reviewer’s insightful perspective, which is valuable for examining the proposed theory and for refining solutions in future research.
> >
> > ## Notation question
> >
> > We thank the reviewer for pointing out this flaw in notation.
> >
> > Given our interest in the probability of correctness, $P(s_i | \alpha_k, M)$ can be further simplified to $P(c_i | \alpha_k, M)$ as suggested by the reviewer. We have fixed this notation in our revised paper.
> >
> > For $s_i$ and $\alpha_k$, they are different in meaning and is necessary for our derivation.
> >
> > * $s_i$ is the final answer (a string) from a single generation $g_i$.
> > * $\mathcal{A} = \{\alpha_1, ..., \alpha_m\}$ is the set of unique answers present in the pool of L generations. For example, if we generate L=10 responses, we might get 7 "A"s and 3 "B"s. Here, $s_1$="A", ..., $s_7$="A", $s_8$="B", etc. The set of unique answers is $\mathcal{A}$ = {$\alpha_1, \alpha_2$} = {"A", "B"}. We have clarified this notation in our revision.
> >
> >
> >
> > If there are no further concerns, we appreciate the reviewer considering raising our score in the evaluation.

---

### Author Response · Authors · 2025-12-01
**Summary of Rebuttal and Discussion**

Dear Area Chair,

We thank the reviewers for their constructive feedback and engagement. During the discussion period, we provided extensive clarifications and new experimental results that we believe have fully resolved the raised concerns. We are pleased to note that **Reviewer suxy and hEsP have both raised their score to 8 *well before* the unfortunate leakage of identities according to [Openreview's Statement](https://openreview.net/forum/user%7Cstatement_regarding_api_security_incident)**. Specifically, Reviewers suxy and hEsP raised their scores on November 24 (15:22 UTC) and November 27 (03:51 UTC), respectively—*well before* the leakage first occurred on November 27 at 15:09 UTC. While Reviewer z2Y8 and rGBo are unable to respond to our rebuttal, we believe their concerns are well-addressed as we summarize below.

## Summary of Strengths

The reviewers unanimously recognized the value of our contribution to Test-Time Scaling (TTS) and the quality of our presentation:

- **Theoretical Clarity:** Reviewers praised the framework as "**intuitive and clear**" (Reviewer suxy) and noted the "**clean mathematical formulation**" (Reviewer rGBo). Reviewer z2Y8 highlighted that the theoretical results regarding PRM score distributions "**could be of independent interest to the research community.**"
- **Empirical Validation:** Reviewers emphasized that "**The experiments are quite comprehensive**" (Reviewer rGBo) and our method "**significantly improves test-time-scaling efficiency**" (Reviewer hEsP) and verified that we "**achieve similar accuracy as baselines while using only 21.3%-37.1% of the computes**" (Reviewer rGBo).
- **Novelty:** Reviewer hEsP commended the finding that "**negative weights are necessary**" in utilizing PRM signals effectively, and Reviewer suxy concluded that the "**contribution of the MAP and negative weights formulation is of interest to the ICLR community.**" Reviewer z2Y8 further noted that our findings regarding PRM score distributions are "**quite interesting and could be of independent interest to the research community**."

## Response to Key Concerns

### 1. Theoretical Validity
Reviewers z2Y8 and rGBo raised questions regarding the theoretical underpinnings of our framework.
- **Clarification on Derivation (Reviewer z2Y8):** Reviewer z2Y8 expressed confusion regarding the Bayes' theorem derivation and the definition of the posterior probability. In our rebuttal, we clarified that these concerns probably arose from a misreading of the formulaic decomposition. We pointed out that our derivation uses the chain rule ($P(\mathcal{G}, \mathcal{P}|...) = P(\mathcal{P}|\mathcal{G},...)P(\mathcal{G}|...)$) to ***explicitly* model the causal dependency of PRM scores on LLM generations, rather than ignoring it**.
- **Clarification on Independence (Reviewer rGBo):** Reviewer rGBo questioned the independence assumption (Assumption 3.1). We clarified that this was a misunderstanding of the Maximum Likelihood Estimation setup; our framework holds because **we condition on the specific model generation distribution and verifier distribution**, rendering the samples conditionally independent.
- **Empirical Validation (Reviewer suxy):** We further validated our theory via an ablation study which confirmed that the performance gap between the theoretical KDE model and the practical Logit method is **strictly due to estimation noise in high-dimensional spaces, not a theoretical flaw**.
### 2. Generalization and Statistical Significance
To address concerns about the experiments being limited to datasets in the math domain (hEsP, rGBo, z2Y8), we:
- Expanded evaluations to **11 domains that vary significantly**, where our method **consistently outperforms all baselines**.
- Demonstrated **cross-domain transferability** (e.g., calibrating on Math and testing on History) with strong results.
- Provided standard deviations to confirm that our performance gains are **statistically significant** (addressing reviewer rGBo and z2Y8), a gain of 2.02 points with a very low standard deviation of 0.02.

### 3. Computational Cost and Efficiency
Reviewer z2Y8 raised concerns regarding the overhead of generating the calibration set. We demonstrated that **the calibration cost is a negligible, one-time investment**. Specifically, the cost of calibrating on just 100 questions is fully recovered after serving approximately 127 user queries. In any practical deployment serving thousands of requests, this break-even point is reached almost immediately, rendering the calibration overhead insignificant relative to the long-term efficiency gains.

We believe we have addressed all concerns regarding theory, cost, and generalization, and that our work stands as a theoretically grounded and practically efficient advancement in Test-Time Scaling. Additionally, all the clarification and additional experiments mentioned are readily incorporated into the revised paper marked in blue for your review.

Sincerely,

The Authors

---

### Meta-Review · Area_Chair_k7vs · 2026-01-07

**Summary:**

The reviewers were split about this paper and did not come to a consensus. On one hand they appreciated the theoretical framing and experiments of the paper. On the other they had issues with (a) the utility of the model, (b) the paper clarity, (c) generalizability beyond the MATH dataset, (d) the independence assumption, (e) the ablation study, (f) missing statistical significance, (g) clarity of the theory. For (a) the authors responded convincingly to the reviewer’s concern and a follow-up question. For (b) the authors added a sentence to the abstract. For (c) the authors evaluated their approaches on the MMLU-Pro-CoT-Eval dataset. For (d) the authors have convinced me that the independence assumption is valid because of conditioning (the last paragraph in the response is the clearest). The authors further respond to (e)-(g) convincingly. As all of the concerns have been responded to, I vote to accept.

**Reviewer Concerns:**

Please see above.

**Reviewer Scores:**

I believe reviewers would have kept or increased their scores. NOTE: The authors do a bit of a fishy thing when they say: "We are pleased to note that Reviewer suxy and hEsP have both raised their score to 8 well before the unfortunate leakage of identities according to Openreview's Statement. Specifically, Reviewers suxy and hEsP raised their scores on November 24 (15:22 UTC) and November 27 (03:51 UTC), respectively—well before the leakage first occurred on November 27 at 15:09 UTC." This is not strictly true: the identification of the leakage occurred on November 27 at 15:09 UTC, the leakage itself happened before this and the Reviewers and Authors could have colluded. I don't have any evidence that there was collusion, but I wanted to flag this.

---

### Decision · Program_Chairs · 2026-01-26

Accept (Poster)